# Hierarchical Decomposition of Prompt-Based Continual Learning: Rethinking Obscured Sub-optimality

**Liyuan Wang[1], Jingyi Xie[1], Xingxing Zhang[1]\*, Mingyi Huang[1], Hang Su[1], Jun Zhu[1]\***

Dept. of Comp. Sci. & Tech., Institute for AI, BNRist Center, THBI Lab,
Tsinghua-Bosch Joint Center for ML, Tsinghua University, Beijing, China.
`wly19@tsinghua.org.cn`, `jingyi_xie96@163.com`, `xxzhang1993@gmail.com`
`huangmingyi2002@126.com`, `{suhangss, dcszj}@tsinghua.edu.cn`

## Abstract

Prompt-based continual learning leverages pre-trained knowledge for downstream continual learning, and has almost reached the performance pinnacle under supervised pre-training. However, our empirical study reveals that the current strategies fall short of their full potential under the more realistic self-supervised pre-training, which is essential for handling vast quantities of unlabeled data in practice. This is largely due to the difficulty of task-specific knowledge being incorporated into instructed representations via prompt parameters and predicted by uninstructed representations at test time. To overcome such sub-optimality, we conduct a theoretical analysis of the continual learning objective in the context of pre-training, and decompose it into hierarchical components: within-task prediction, task-identity inference, and task-adaptive prediction. Based on these empirical and theoretical insights, we propose Hierarchical Decomposition (HiDe-)Prompt, an innovative approach that explicitly optimizes the hierarchical components with an ensemble of task-specific prompts and statistics of both uninstructed and instructed representations, further with the coordination of a contrastive regularization strategy. Our extensive experiments demonstrate the superior performance of HiDe-Prompt and its robustness to pre-training paradigms in continual learning (e.g., up to 15.01% and 9.61% lead on Split CIFAR-100 and Split ImageNet-R, respectively). Our code is available at `https://github.com/thu-ml/HiDe-Prompt`.

## 1 Introduction

In the realm of artificial intelligence, continual learning has become an area of significant interest. One of the pivotal techniques that greatly facilitate this domain is pre-training, which not only delivers positive knowledge transfer but also enhances resilience to catastrophic forgetting [27, 23, 37, 26, 43]. A recent innovation is the implementation of prompt-based methodologies, which freeze a pre-trained transformer backbone and employ a few prompt parameters to steer representation learning. Such approaches typically involve *construction* of adaptive prompts for each task and *inference* of appropriate prompts during the test phase. By exploring prompt architectures to accommodate task-sharing and task-specific knowledge, this emerging direction demonstrates distinct superiority, almost reaching the upper bound of continual learning performance under supervised pre-training.

Nonetheless, given that robust pre-trained models typically necessitate the learning of substantial amounts of unlabeled data in a self-supervised manner, the influence of pre-training paradigms

---

\*Corresponding authors.

37th Conference on Neural Information Processing Systems (NeurIPS 2023).

on the effectiveness of prompt-based continual learning represents a significant and unresolved query. To answer this question, we first perform an extensive empirical investigation, and it clearly demonstrates the sub-optimality of recent prompt-based approaches under the more realistic self-supervised pre-training. Since self-supervised representations tend to be more general, task-specific knowledge is difficult to incorporate into instructed representations via prompt parameters, as well as predicted by uninstructed representations at test time. Consequently, the performance of many recent approaches, such as L2P [41], DualPrompt [40], S-Prompt [39] and CODA-Prompt [30], is seriously compromised. We further disclose the importance of adaptive prediction for all tasks together, which can potentially mitigate the aforementioned shortcomings to some extent.

Motivated by these observations, we provide an in-depth theoretical analysis of the continual learning objective in the context of pre-training, which can be decomposed into hierarchical components such as *within-task prediction*, *task-identity inference* and *task-adaptive prediction*. Thanks to the well-distributed representations resulting from adequate pre-training, the hierarchical components can be optimized explicitly by constructing an ensemble of task-specific prompts and exploiting the preserved statistics of uninstructed and instructed representations. A novel contrastive regularization is further devised to coordinate these hierarchical components. We refer to this approach as Hierarchical Decomposition (HiDe-)Prompt and demonstrate its superiority through extensive continual learning experiments, especially under the more realistic self-supervised pre-training.

Our contributions include: (1) We provide an extensive empirical study under self-supervised pre-training to demonstrate the sub-optimality of current progress in prompt-based continual learning; (2) To overcome such sub-optimality, we theoretically analyze the objective of continual learning with pre-training, and decompose it into hierarchical components for model design; (3) With task-specific prompts and representation statistics, we propose an innovative approach to optimize the hierarchical components explicitly; (4) Across various continual learning benchmarks and pre-training paradigms, our approach achieves clearly state-of-the-art performance in a rehearsal-free manner.

## 2   Related Work

**Continual Learning:** The ability of continual learning is critical for artificial neural networks to accommodate real-world changes [37, 35]. Numerous efforts in this direction have been devoted to overcoming catastrophic forgetting [22, 34, 33]. According to a recent survey [37], representative strategies include selective stabilization of network parameters, replay of a few old training samples, explicit manipulation of optimization programs, exploitation of well-distributed representations, construction of task-specific parameters, etc. The performance of such strategies varies with particular settings of continual learning. As one of the most challenging and representative settings, class-incremental learning (CIL) [31, 37] requires a continual learning model to perform all old tasks (or classes) without the oracle of task identity. Strong CIL methods generally depend on storage and rehearsal of old training samples [28, 8, 38], which result in efficiency and privacy issues.

**Self-Supervised Learning and Pre-Training:** The exploitation of well-distributed representations, especially from the success of large-scale pre-training, brings significant benefits for downstream continual learning [37, 27, 23]. Due to the scarcity and expense of explicit labeling in many real-world applications, self-supervised learning is typically involved in the pre-training stage to cope with huge amounts of unlabeled data. In particular, instance discrimination [4, 7] with contrastive learning [25] has become the dominant strategy, which aims to maximize representation similarity of the same instance and minimize representation similarity of different instances. Besides, self-supervised paradigms have been shown less sensitive to catastrophic forgetting in upstream continual learning [9], providing a practical way to enrich pre-trained knowledge from in-the-wild data.

**Prompt-Based Approach:** Inspired by parameter-efficient fine-tuning techniques in NLP [11, 10], recent prompt-based approaches [3, 41, 40, 39, 30] are developed to leverage pre-trained knowledge adaptively for downstream continual learning. The basic idea includes *construction* and *inference* of adaptive prompts for each task, so as to instruct a frozen transformer backbone. The former mainly focuses on exploring prompt architectures to instruct representations with task-sharing and task-specific knowledge, closely related to the discussion of model architectures in continual learning [37, 36], while the latter attempts to predict appropriate (combinations of) prompts with uninstructed representations. Although such methods have achieved remarkably strong performance under supervised pre-training, whether these advantages are consistent under the more realistic self-

supervised pre-training remains to be explored. A concurrent study [43] observed that self-supervised pre-training is more challenging for continual learning approaches that require fine-tuning of the backbone, implying a non-trivial impact of pre-training paradigms on downstream continual learning.

# 3 Preliminary Analysis

In this section, we first introduce the problem formulation of prompt-based continual learning, and then evaluate the impact of pre-training paradigms with an extensive empirical study.

## 3.1 Formulation of Prompt-Based Continual Learning

**Continual learning** aims to learn a sequence of tasks on their respective training sets $\mathcal{D}_1, ..., \mathcal{D}_T$ and excel on their corresponding test sets. The training set for task $t$ typically consists of various data-label pairs $\mathcal{D}_t = \{(\boldsymbol{x}_{t,n}, y_{t,n})\}_{n=1}^{N_t}$, where $\boldsymbol{x}_{t,n} \in \mathcal{X}_t$ and $y_{t,n} \in \mathcal{Y}_t$ represent the sample and label elements, respectively. We will use $|\cdot|$ to denote the cardinality of a set and $[N] = \{1, 2, \cdots, N\}$ as the set of intergers from 1 to $N$. Consider a neural network model with a backbone $f_\theta$ parameterized by $\theta$, and an output layer $h_\psi$ parameterized by $\psi$. This model seeks to learn the projection from $\mathcal{X} = \bigcup_{t=1}^{T} \mathcal{X}_t$ to $\mathcal{Y} = \bigcup_{t=1}^{T} \mathcal{Y}_t$, aiming to predict the label $y = h_\psi(f_\theta(\boldsymbol{x})) \in \mathcal{Y}$ of an unseen test sample $\boldsymbol{x}$ drawn from previous tasks. The backbone function $f_\theta$ is assumed to be pre-trained with a substantial quantity of additional training samples external to each $\mathcal{D}_t$. There are commonly three distinct settings for continual learning [31]: task-, domain-, and class-incremental learning (TIL, DIL, and CIL). Specifically, $\mathcal{Y}_1, ..., \mathcal{Y}_T$ are identical for DIL while disjoint for TIL and CIL. The task identity is provided for TIL at test time but is not available for DIL and CIL. Therefore, CIL is considered to be more representative and challenging in general. Of note, the continual learning process is *rehearsal-free* [30]—all elements of $\mathcal{D}_t$ are available only when learning task $t$.

**Prompt-based approaches** for vision tasks further specify the backbone $f_\theta$ as a pre-trained vision transformer (ViT), where multiple consecutive multi-head self-attention (MSA) layers can transform an input sample into a sequence-like output representation $\boldsymbol{h} \in \mathbb{R}^{L_h \times D}$ of sequence length $L_h$ and embedding dimension $D$. The backbone parameters $\theta$ are typically frozen to obtain generalizable representations. A few prompt parameters $\boldsymbol{p} \in \mathbb{R}^{L_p \times D}$ of sequence length $L_p$ and embedding dimension $D$ are prepended to $\boldsymbol{h}$ to exploit the pre-trained knowledge adaptively. Here we denote the input of the $l$-th MSA layer as $\boldsymbol{h}^l \in \mathbb{R}^{L_{h^l} \times D}$, which consists of query $\boldsymbol{h}_Q^l$, key $\boldsymbol{h}_K^l$ and value $\boldsymbol{h}_V^l$, and denote the prompt as $\boldsymbol{p}^l \in \mathbb{R}^{L_{p^l} \times D}$. For notation clarity, we take one MSA layer as an example and omit the layer label $l$ if not necessary. Then, the output of this MSA layer is given as

$$\text{MSA}(\boldsymbol{h}_Q, \boldsymbol{h}_K, \boldsymbol{h}_V) = \text{Concat}(h_1, ..., h_m)W_O, \tag{1}$$

$$h_i = \text{Attention}(\boldsymbol{h}_Q W_{Q,i}, \boldsymbol{h}_K W_{K,i}, \boldsymbol{h}_V W_{V,i}), i \in [m], \tag{2}$$

where $W_O$, $W_{Q,i}$, $W_{K,i}$ and $W_{V,i}$ are projection matrices, $m$ is the number of heads, and $\boldsymbol{h}_Q = \boldsymbol{h}_K = \boldsymbol{h}_V$ in ViT. There are two major implementations of prompt-based methodologies [40], i.e., Prompt Tuning (ProT) [18] and Prefix Tuning (PreT) [19]. Specifically, ProT prepends an identical $\boldsymbol{p}$ to $\boldsymbol{h}_Q$, $\boldsymbol{h}_K$ and $\boldsymbol{h}_V$:

$$f_{\text{ProT}}(\boldsymbol{p}, \boldsymbol{h}) = \text{MSA}([\boldsymbol{p}; \boldsymbol{h}_Q], [\boldsymbol{p}; \boldsymbol{h}_K], [\boldsymbol{p}; \boldsymbol{h}_V]), \tag{3}$$

where $[\cdot ; \cdot]$ denotes the concatenation operation along the dimension of sequence length, and the output in $\mathbb{R}^{(L_h + L_p) \times D}$ has increased dimensions. In contrast, PreT splits $\boldsymbol{p}$ into $\boldsymbol{p}_K \in \mathbb{R}^{L_p/2 \times D}$ and $\boldsymbol{p}_V \in \mathbb{R}^{L_p/2 \times D}$ only for $\boldsymbol{h}_K$ and $\boldsymbol{h}_V$, respectively:

$$f_{\text{PreT}}(\boldsymbol{p}, \boldsymbol{h}) = \text{MSA}(\boldsymbol{h}_Q, [\boldsymbol{p}_K; \boldsymbol{h}_K], [\boldsymbol{p}_V; \boldsymbol{h}_V]), \tag{4}$$

where the output dimension remains the same as the input $\boldsymbol{h} \in \mathbb{R}^{L_h \times D}$. As the training samples for each task are introduced sequentially, prompt-based continual learning needs to incorporate task-specific knowledge into prompt parameters while overcoming their catastrophic forgetting. The mainstream idea is to construct adaptive prompts for each task and then infer appropriate (combinations of) prompts at test time. Here we compare state-of-the-art approaches from these two aspects, demonstrated conceptually in Fig. 1:

**L2P** [41] constructs a prompt pool $\boldsymbol{P} = \{\boldsymbol{p}_1, ..., \boldsymbol{p}_M\}$ potentially shared by all tasks where $M$ is the total number of prompts, and then instructs the last MSA layer in a ProT fashion. Each prompt $\boldsymbol{p}_i$ is associated to a learnable key $\boldsymbol{k}_i \in \mathbb{R}^D$, optimized by the cosine distance $\gamma(q(\boldsymbol{x}), \boldsymbol{k}_i)$ of the top-$N$ keys to a query function $q(\boldsymbol{x}) = f_\theta(\boldsymbol{x})[0]$ to incorporate knowledge. Therefore, the most relevant keys and the corresponding prompts can be selected with uninstructed representations for inference.

**DualPrompt** [40] constructs task-sharing prompts $g^l$ and task-specific prompts $e_t^l$ to instruct different MSA layers in a PreT fashion. All $e_t^l$ belonging to the same task is associated to a task-specific key $k_t \in \mathbb{R}^D$, optimized by $\gamma(q(x), k_t)$, and the index of the best-matched key is selected for inference.

**S-Prompt** [39] constructs only task-specific prompts $e_t$ for each task, and adopts a similar ProT strategy as L2P to instruct the last MSA layer. The inference of task identity is achieved by a simple KNN

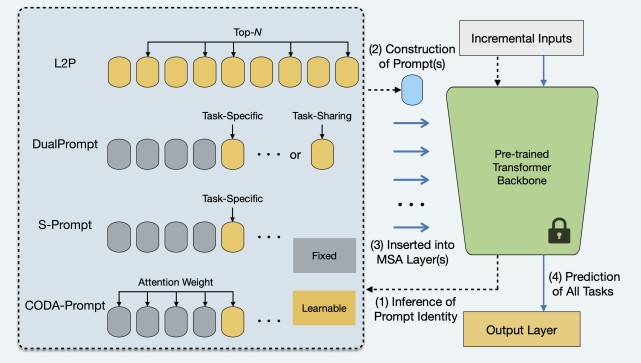

Figure 1: Illustration of prompt-based continual learning.

strategy for the nearest task centroid. Unlike other methods, S-Prompt associates an exclusive output head $\psi_t$ to each task $t = 1, ..., T$.

**CODA-Prompt** [30] exploits the prompt pool $P$ by its weighted summation, i.e., $p = \sum_{i=1}^{M} \alpha_i p_i$, where $\alpha_i = \gamma(q(x), k_i)$ is the weighting factor, and adopts a similar PreT strategy as DualPrompt to instruct multiple MSA layers. The inference of $\alpha_i$ enables construction of adaptive prompts.

### 3.2 Empirical Study of Pre-Training Paradigms

Either explicitly or implicitly, the above prompt-based approaches all incorporate the knowledge of each task into prompt parameters and predict their identities from uninstructed representations. To evaluate the impact of pre-training paradigms, we perform an empirical study with widely-used CIL benchmarks such as Split CIFAR-100 and Split ImageNet-R [41, 40]. In addition to supervised pre-training of ImageNet-21K [29] (denoted as Sup-21K), we consider several powerful self-supervised models that release ViT checkpoints[2], such as iBOT [44], DINO [2] and MoCo v3 [5].

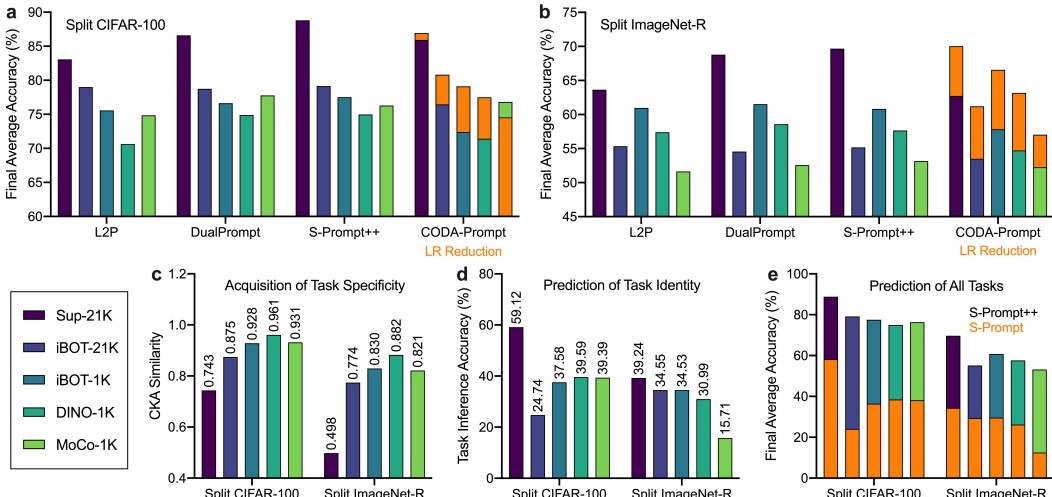

Figure 2: Empirical study of prompt-based continual learning under different pre-training paradigms.

We carefully evaluate the official implementations of all baselines for fair comparison. We follow largely the training regimes of L2P [41] and DualPrompt [40], which are basically consistent. As S-Prompt [39] is initially designed for DIL, we slightly modify its implementation by inserting task-specific prompts into the same layers as DualPrompt (i.e., layers 1-5) in a PreT manner, so as to evaluate the impact of prompt architectures. The output layer retains multiple heads associated

---

[2]iBOT currently achieves the first-place performance for self-supervised classification on ImageNet and releases checkpoints on both ImageNet-21K and -1K, while others only release checkpoints on ImageNet-1K.

with the task identity (still denoted as S-Prompt), or a single head as with other baselines (denoted as S-Prompt++). CODA-Prompt [30] is officially implemented in a DualPrompt-like architecture but depends heavily on the use of a smaller learning rate with cosine decay. Here we present its performance with both default and reduced learning rates. Using the same learning rate as [41, 39], we grid search for an appropriate number of epochs (detailed in Appendix D) and report the best performance for all baselines.

As shown in Fig. 2, a, b, the above prompt-based approaches achieve outstanding performance under Sup-21K, where the use of task-specific prompts clearly outperforms task-sharing prompts (i.e., S-Prompt++ ≈ CODA-Prompt > DualPrompt > L2P) due to explicit avoidance of catastrophic forgetting. However, the four baselines suffer **significant performance degradation** under the more realistic self-supervised pre-training. In particular, the performance differences between prompt architectures have become much smaller, suggesting that the task-specific and task-sharing knowledge are not well differentiated. Besides, CODA-Prompt can generally achieve leading performance as a direct result of the learning rate (LR) reduction rather than the prompt architecture.

We perform two additional experiments to demonstrate the **obscured sub-optimality**[3]. First, we evaluate the CKA similarity of uninstructed and instructed representations by learning task-specific prompts (Fig. 2, c). The CKA similarity of self-supervised pre-training is significantly higher, suggesting a greater difficulty for prompt parameters to incorporate task-specific knowledge. Second, we evaluate the ability to predict task identity from uninstructed representations and task-specific keys, where self-supervised pre-training exhibits much lower accuracy (Fig. 2, d). Interestingly, although only less than 40% task identities are correctly predicted, S-Prompt++ can still achieve considerable (albeit sub-optimal) performance, owing to the compensation effects of using a single-head output layer (Fig. 2, e). Together with the results in Fig. 2, c, d, it is conceivable that using an "incorrect" prompt would not severely affect the instructed representations, which can still be correctly predicted in a well-balanced single-head output layer. In contrast, S-Prompt performs much worse than S-Prompt++, as its multi-head output layer undertakes all errors of task-identity inference.

## 4  Theoretical Foundation and Our Approach

In this section, we first present a theoretical analysis of the sufficient and necessary conditions for improving continual learning in the context of pre-training, and then present an innovative approach for prompt-based continual learning to achieve this objective.

### 4.1  Hierarchical Decomposition of Continual Learning Objective

For continual learning of sequentially arrived $\mathcal{D}_t$, $\mathcal{X}_t$ and $\mathcal{Y}_t$ are the domain and label of task $t$. Here we take CIL as a typical scenario for theoretical analysis where $\mathcal{Y}_t \cap \mathcal{Y}_{t'} = \emptyset$, $\forall t \neq t'$ (see Appendix A for DIL and TIL). Let $\mathcal{X}_t = \bigcup_j \mathcal{X}_{t,j}$ and $\mathcal{Y}_t = \{\mathcal{Y}_{t,j}\}$, where $j \in [|\mathcal{Y}_t|]$ indicates the $j$-th class in task $t$. Now assume we have a ground event denoted as $\mathcal{D} = \{\mathcal{D}_1, ..., \mathcal{D}_t\}$ and a pre-trained model $f_\theta$. For any sample $\boldsymbol{x} \in \bigcup_{k=1}^t \mathcal{X}_k$, a general goal of the CIL problem is to learn $P(\boldsymbol{x} \in \mathcal{X}_{i,j}|\mathcal{D}, \theta)$, where $i \in [t]$ and $j \in [|\mathcal{Y}_i|]$. This can be decomposed into two probabilities, including task-identity inference (TII) and within-task prediction (WTP), denoted as $P(\boldsymbol{x} \in \mathcal{X}_i|\mathcal{D}, \theta)$ and $P(\boldsymbol{x} \in \mathcal{X}_{i,j}|\boldsymbol{x} \in \mathcal{X}_i, \mathcal{D}, \theta)$, respectively. Based on Bayes' theorem, we have

$$P(\boldsymbol{x} \in \mathcal{X}_{i,j}|\mathcal{D}, \theta) = P(\boldsymbol{x} \in \mathcal{X}_{i,j}|\boldsymbol{x} \in \mathcal{X}_i, \mathcal{D}, \theta)P(\boldsymbol{x} \in \mathcal{X}_i|\mathcal{D}, \theta). \tag{5}$$

Let $\bar{i} \in [t]$ and $\bar{j} \in [|\mathcal{Y}_i|]$ be the ground truth of an $\boldsymbol{x}$ w.r.t. the task identity and within-task index. Eq. (5) shows that if we can improve either the WTP performance $P(\boldsymbol{x} \in \mathcal{X}_{\bar{i},\bar{j}}|\boldsymbol{x} \in \mathcal{X}_{\bar{i}}, \mathcal{D}, \theta)$, the TII performance $P(\boldsymbol{x} \in \mathcal{X}_{\bar{i}}|\mathcal{D}, \theta)$, or both, then the CIL performance $P(\boldsymbol{x} \in \mathcal{X}_{\bar{i},\bar{j}}|\mathcal{D}, \theta)$ would be improved. However, such an improvement is limited since it is upper-bounded by WTP or TII. To further improve the CIL performance, we propose a hierarchical decomposition of its objective. That is, besides the improvement of $P(\boldsymbol{x} \in \mathcal{X}_{\bar{i},\bar{j}}|\mathcal{D}, \theta)$, we also need to improve the performance of task-adaptive prediction (TAP), denoted as $P(\boldsymbol{x} \in \mathcal{X}^y|\mathcal{D}, \theta)$, where $\mathcal{X}^y$ represents the domain of

---

[3]The experiments in Fig. 2, c, d employ the same prompt architecture and inference methodology as S-Prompt++ to explicitly demonstrate the impact of task specificity. In fact, the objective of continual learning is tantamount to optimizing the probability distribution on the left side of Eq. (5), which can be decomposed into the two probabilities on the right side corresponding to Fig. 2, c, d, respectively. Therefore, such empirical analysis is representative for prompt-based continual learning from a theoretical perspective.

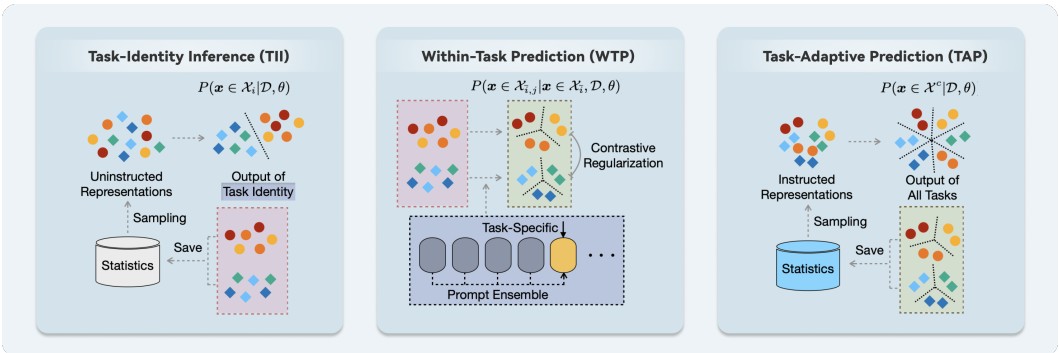

Figure 3: Illustration of Hierarchical Decomposition (HiDe-)Prompt.

class $y$ in all previous tasks, and $y = \mathcal{Y}_{\bar{i},\bar{j}}$ is the ground truth label of $\boldsymbol{x}$. Then the final goal of CIL is formulated as a multi-objective optimization problem, i.e., $\max[P(\boldsymbol{x} \in \mathcal{X}_{\bar{i},\bar{j}}|\mathcal{D}, \theta), P(\boldsymbol{x} \in \mathcal{X}^y|\mathcal{D}, \theta)]$. Notice that the TII probability is a categorical distribution over all observed tasks upto $t$, while the TAP probability is over all observed classes $\bigcup_{k=1}^{t} \mathcal{Y}_k$.

To resolve the problems above, we derive the sufficient and necessary conditions in the context of the widely-used cross-entropy loss. Specifically, we define

$$H_{\text{WTP}}(\boldsymbol{x}) = \mathcal{H}(\mathbf{1}_{\bar{j}}, \{P(\boldsymbol{x} \in \mathcal{X}_{i,j}|\boldsymbol{x} \in \mathcal{X}_i, \mathcal{D}, \theta)\}_j), \tag{6}$$

$$H_{\text{TII}}(\boldsymbol{x}) = \mathcal{H}(\mathbf{1}_{\bar{i}}, \{P(\boldsymbol{x} \in \mathcal{X}_i|\mathcal{D}, \theta)\}_i), \tag{7}$$

$$H_{\text{TAP}}(\boldsymbol{x}) = \mathcal{H}(\mathbf{1}_{\bar{c}}, \{P(\boldsymbol{x} \in \mathcal{X}^c|\mathcal{D}, \theta)\}_c), \tag{8}$$

where $H_{\text{WTP}}$, $H_{\text{TII}}$, and $H_{\text{TAP}}$ are the cross-entropy values of WTP, TII, and TAP, respectively. The operation $\mathcal{H}(p, q) \triangleq -\mathbb{E}_p[\log q] = -\sum_i p_i \log q_i$. $\mathbf{1}$. is a one-hot encoding function.

We now present the first theorem under the CIL scenario (see Appendix A for a detailed proof):

**Theorem 1** *For continual learning with pre-training, if $\mathbb{E}_{\boldsymbol{x}}[H_{\text{WTP}}(\boldsymbol{x})] \leq \delta$, $\mathbb{E}_{\boldsymbol{x}}[H_{\text{TII}}(\boldsymbol{x})] \leq \epsilon$, and $\mathbb{E}_{\boldsymbol{x}}[H_{\text{TAP}}(\boldsymbol{x})] \leq \eta$, we have the loss error $\mathcal{L} \in [0, \max\{\delta + \epsilon, \eta\}]$, regardless whether WTP, TII and TAP are trained together or separately.*

With the use of cross-entropy, the continual learning performance tends to be better as the bounds are tightened. In Theorem 1 we have shown that good performances of WTP, TII and TAP are sufficient to guarantee a good performance of CIL. For completeness, we now study the necessary conditions of a well-performed CIL model in Theorem 2.

**Theorem 2** *For continual learning with pre-training, if the loss error $\mathcal{L} \leq \xi$, then there always exist (1) a WTP, s.t. $H_{\text{WTP}} \leq \xi$; (2) a TII, s.t. $H_{\text{TII}} \leq \xi$; and (3) a TAP, s.t. $H_{\text{TAP}} \leq \xi$.*

Theorem 2 suggests that if a continual learning model is well trained (i.e., with low loss), then the WTP, TII and TAP for sequential tasks are always implied to be small. It is worth noting that without the pre-trained knowledge carried by $\theta$, Theorem 1 and Theorem 2 would degrade to the main conclusion of a previous theoretical study [13], suggesting that the presented theorems are particularly directed to the impact of *pre-training* for continual learning (detailed in Appendix B). Besides, the paradigm of pre-training is indeed related to the performance of continual learning, because it can affect the distribution of representations from $f_\theta$, and further WTP, TII and TAP. Previous work has demonstrated that self-supervised representations tend to be more robust to parameter changes than supervised ones [9, 21, 37], which is beneficial for accumulating pre-trained knowledge (if applicable) but challenging for adapting to downstream tasks on a continual basis (see Fig. 2, c, d).

## 4.2 HiDe-Prompt for Prompt-Based Continual Learning

Motivated by the above empirical and theoretical insights, we propose to optimize explicitly the hierarchical components (i.e., WTP, TII and TAP) for prompt-based continual learning, as shown in Fig. 3. Our proposal stems from a particular advantage of pre-training, where the distributions of uninstructed and instructed representations can be effectively preserved through their statistical information. In the case of classification, for example, since each class tends to have single-peaked

representations (see Appendix D, Fig. 5 and Fig. 6), we can naturally approximate them with Gaussian distributions. For generality, here we denote the approximated distributions of uninstructed and instructed representations as $\hat{\mathcal{G}}_c$ and $\mathcal{G}_c$ for each class $c \in \mathcal{Y}_i, i \in [t-1]$, respectively, and discuss their specific forms latter.

First, we improve **WTP** through effective incorporation of task-specific knowledge. We construct an expandable prompt pool with only task-specific prompts $e_t$ to incorporate the knowledge of $\mathcal{D}_t$, optimized by a cross-entropy (CE) loss for $H_{\mathrm{WTP}}$. The previous prompts $e_1, ..., e_{t-1}$ are frozen to avoid catastrophic forgetting. In order to transfer knowledge for learning each task effectively, we employ a *prompt ensemble* (PE) strategy, where the current prompt is initialized by the last prompt $e_t \leftarrow e_{t-1}$ and then optimized with a weighted combination of all previous prompts $p_t = \alpha \sum_{i=1}^{t-1} e_i + (1-\alpha)e_t$. $\alpha$ is a hyperparameter that controls the strength of inherited old knowledge to facilitate $p_t$ in learning the current task. Meanwhile, the instructed representations of $p_t$, although allowing the new task to be performed well, may overlap with that of the old tasks and thus affect TAP. To overcome this issue, we exploit the old-task statistics of instructed representations (collected by $f_\theta$ and $p_i$ for $i = 1, ..., t-1$), where for classification we calculate the mean $\mu_c$ of $\mathcal{G}_c$ for each class $c \in \mathcal{Y}_i$, and design a *contrastive regularization* (CR):

$$\mathcal{L}_{\mathrm{CR}}(p_t) = \sum_{h \in \mathcal{H}_t} \frac{1}{\sum_{i=1}^{t-1} |\mathcal{Y}_i|} \sum_{i=1}^{t-1} \sum_{c \in \mathcal{Y}_i} \log \frac{\exp(h \cdot \mu_c / \tau)}{\sum_{h' \in \mathcal{H}_t} \exp(h \cdot h' / \tau) + \sum_{i=1}^{t-1} \sum_{c \in \mathcal{Y}_i} \exp(h \cdot \mu_c / \tau)}, \quad (9)$$

where $\mathcal{H}_t$ is the embedding transformation of $\mathcal{D}_t$ with $f_\theta$ and $p_t$. $\tau$ is the temperature coefficient, which is insensitive and set to 0.8 in practice. Notably, here we use only $\mu_c$ to represent each class for efficiency, which can be optionally replaced by sampling from $\mathcal{G}_c$ for better performance.

Then, the loss function of WTP can be defined as

$$\mathcal{L}_{\mathrm{WTP}}(\psi, p_t) = \mathcal{L}_{\mathrm{CE}}(\psi, p_t) + \lambda \mathcal{L}_{\mathrm{CR}}(p_t). \quad (10)$$

Therefore, the instructed representations of new classes can be well distinguished for WTP while avoiding overlap with the previous ones. $\lambda$ is a hyperparamter to balance the impact of old classes.

Second, we improve **TII** and **TAP** through exploiting the approximated distributions of uninstructed and instructed representations, respectively. For TII, we construct an auxiliary output layer $\hat{h}_\omega : \mathbb{R}^D \to \mathbb{R}^T$ parameterized by $\omega$, learning explicitly the projection from uninstructed representations to task identity via cross-entropy (i.e., $H_{\mathrm{TII}}$):

$$\mathcal{L}_{\mathrm{TII}}(\omega) = \frac{1}{\sum_{i=1}^{t} |\mathcal{Y}_i|} \sum_{i=1}^{t} \sum_{c \in \mathcal{Y}_i} \sum_{\hat{h} \in \hat{\mathcal{H}}_{i,c}} -\log \frac{\exp(\hat{h}_\omega(\hat{h})[i])}{\sum_{j=1}^{t} \exp(\hat{h}_\omega(\hat{h})[j])}, \quad (11)$$

where $\hat{\mathcal{H}}_{i,c}$ is constructed by sampling an equal number of pseudo representations from $\hat{\mathcal{G}}_c$ for $c \in \mathcal{Y}_i$ and $i \in [t]$. Unlike other baselines that freeze the projection of old tasks (i.e., the previous keys), our $\hat{h}_\omega$ is continually adapted for all tasks and thus greatly facilitates TII.

Similarly, the final output layer $h_\psi : \mathbb{R}^D \to \mathbb{R}^{|\mathcal{Y}|}$ can be further optimized for TAP (i.e., $H_{\mathrm{TAP}}$):

$$\mathcal{L}_{\mathrm{TAP}}(\psi) = \frac{1}{\sum_{i=1}^{t} |\mathcal{Y}_i|} \sum_{i=1}^{t} \sum_{c \in \mathcal{Y}_i} \sum_{h \in \mathcal{H}_{i,c}} -\log \frac{\exp(h_\psi(h)[c])}{\sum_{j=1}^{t} \sum_{c' \in \mathcal{Y}_j} \exp(h_\psi(h)[c'])}, \quad (12)$$

where $\mathcal{H}_{i,c}$ is constructed by sampling an equal number of pseudo representations from $\mathcal{G}_c$ for $c \in \mathcal{Y}_i$ and $i = 1, ..., t$. As $\omega$ and $\psi$ are usually *light-weight*, the optimization of TII and TAP is computationally efficient. At test time, HiDe-Prompt predicts the task identity $i = \hat{h}_\omega(f_\theta(x))$ and then the label $y = h_\psi(f_\theta(x; p_i))$. Please refer to Appendix Algorithm 1 for more details.

Since the pre-trained representations are usually well-distributed, there are many reasonable strategies to model $\hat{\mathcal{G}}_c$ and $\mathcal{G}_c$. On default, the distributions of uninstructed and instruted representations can be faithfully recovered by modeling each class as a Gaussian with a dedicated mean and covariance. With adequate pre-training, the covariance can be further reduced to variance for efficiency. Alternatively, such statistical modeling can employ multiple centroids obtained from KNN and add Gaussian noise, which is also an efficient choice and is applicable to other task types.

## 5 Experiment

In this section, we first describe the experimental setups, and then present the experimental results.

Table 1: Overall performance of continual learning. We present the final average accuracy (FAA), cumulative average accuracy (CAA) and final forgetting measure (FFM) with $\pm$ standard deviation under different pre-trained models (PTM), over three runs of different random seeds and task splits. *The original paper [30] used a different supervised checkpoint and a different data split of ImageNet-R. We asked the authors for official code and reproduced its results under the same setting as [41, 40].

| PTM | Method | Split CIFAR-100 | | | Split ImageNet-R | | |
|---|---|---|---|---|---|---|---|
| | | FAA (↑) | CAA (↑) | FFM (↓) | FAA (↑) | CAA (↑) | FFM (↓) |
| Sup-21K | L2P [41] | 83.06 ±0.17 | 88.25 ±0.01 | 6.58 ±0.40 | 63.65 ±0.12 | 67.25 ±0.02 | 7.51 ±0.17 |
| | DualPrompt [40] | 86.60 ±0.19 | 90.64 ±0.01 | 4.45 ±0.16 | 68.79 ±0.31 | 71.96 ±0.04 | 4.49 ±0.14 |
| | S-Prompt++ [39] | 88.81 ±0.18 | 92.25 ±0.03 | 3.87 ±0.05 | 69.68 ±0.12 | 72.50 ±0.04 | 3.29 ±0.05 |
| | CODA-Prompt [30]* | 86.94 ±0.63 | 91.57 ±0.75 | 4.04 ±0.18 | 70.03 ±0.47 | 74.26 ±0.24 | 5.17 ±0.22 |
| | HiDe-Prompt (Ours) | **92.61** ±0.28 | **94.03** ±0.01 | **3.16** ±0.10 | **75.06** ±0.12 | **76.60** ±0.01 | **2.17** ±0.19 |
| iBOT-21K | L2P [41] | 79.00 ±0.28 | 85.13 ±0.05 | 5.55 ±0.36 | 55.35 ±0.28 | 58.62 ±0.05 | 3.73 ±0.53 |
| | DualPrompt [40] | 78.76 ±0.23 | 86.16 ±0.02 | 9.84 ±0.24 | 54.55 ±0.53 | 58.69 ±0.01 | 5.38 ±0.70 |
| | S-Prompt++ [39] | 79.14 ±0.65 | 85.85 ±0.17 | 9.17 ±1.33 | 55.16 ±0.83 | 58.48 ±0.18 | 4.07 ±0.16 |
| | CODA-Prompt [30] | 80.83 ±0.27 | 87.02 ±0.20 | 7.50 ±0.25 | 61.22 ±0.35 | 66.76 ±0.37 | 9.66 ±0.20 |
| | HiDe-Prompt (Ours) | **93.02** ±0.15 | **94.56** ±0.05 | **1.33** ±0.24 | **70.83** ±0.17 | **73.23** ±0.08 | **2.46** ±0.21 |
| iBOT-1K | L2P [41] | 75.57 ±0.41 | 82.69 ±0.06 | 7.23 ±0.93 | 60.97 ±0.26 | 65.95 ±0.02 | 4.07 ±0.66 |
| | DualPrompt [40] | 76.63 ±0.05 | 85.08 ±0.12 | 8.41 ±0.40 | 61.51 ±1.05 | 67.11 ±0.08 | 5.02 ±0.52 |
| | S-Prompt++ [39] | 77.53 ±0.56 | 85.66 ±0.16 | 8.07 ±0.97 | 60.82 ±0.68 | 66.03 ±0.91 | 4.16 ±0.14 |
| | CODA-Prompt [30] | 79.11 ±1.02 | 86.21 ±0.49 | 7.69 ±1.57 | 66.56 ±0.68 | 73.14 ±0.57 | 7.22 ±0.38 |
| | HiDe-Prompt (Ours) | **93.48** ±0.11 | **95.02** ±0.01 | **1.00** ±0.24 | **71.33** ±0.21 | **73.62** ±0.13 | **2.79** ±0.26 |
| DINO-1K | L2P [41] | 70.65 ±0.57 | 79.02 ±0.01 | 9.46 ±1.68 | 57.40 ±0.23 | 62.56 ±0.20 | 3.58 ±0.28 |
| | DualPrompt [40] | 74.90 ±0.21 | 83.98 ±0.16 | 10.26 ±0.62 | 58.57 ±0.45 | 64.89 ±0.15 | 5.80 ±0.21 |
| | S-Prompt++ [39] | 74.97 ±0.46 | 83.82 ±0.39 | 7.78 ±0.66 | 57.64 ±0.16 | 63.79 ±0.05 | 5.08 ±0.31 |
| | CODA-Prompt [30] | 77.50 ±0.64 | 84.81 ±0.30 | 8.10 ±0.01 | 63.15 ±0.39 | 69.73 ±0.25 | 6.86 ±0.11 |
| | HiDe-Prompt (Ours) | **92.51** ±0.11 | **94.25** ±0.01 | **0.99** ±0.21 | **68.11** ±0.18 | **71.70** ±0.01 | **3.11** ±0.17 |
| MoCo-1K | L2P [41] | 74.85 ±0.28 | 83.14 ±0.03 | 6.51 ±0.95 | 51.64 ±0.19 | 58.87 ±0.24 | **2.37** ±0.59 |
| | DualPrompt [40] | 77.77 ±0.68 | 85.31 ±0.07 | 6.61 ±1.08 | 52.57 ±0.82 | 60.65 ±0.16 | 2.73 ±0.49 |
| | S-Prompt++ [39] | 76.30 ±0.54 | 83.88 ±0.12 | 14.67 ±0.64 | 53.15 ±1.10 | 60.03 ±0.95 | 4.11 ±1.84 |
| | CODA-Prompt [30] | 76.83 ±0.34 | 84.97 ±0.23 | 12.60 ±0.02 | 55.75 ±0.26 | 65.49 ±0.36 | 10.46 ±0.04 |
| | HiDe-Prompt (Ours) | **91.57** ±0.20 | **93.70** ±0.01 | **1.19** ±0.18 | **63.77** ±0.49 | **68.26** ±0.01 | 3.57 ±0.96 |

**Benchmark:** We consider multiple CIL benchmarks that are widely used for prompt-based continual learning [41, 40, 30]. Specifically, Split CIFAR-100 [14] includes 100-class small-scale images, randomly split into 10 incremental tasks of disjoint classes. Split ImageNet-R [14] includes 200-class large-scale images that are hard examples of ImageNet [29] or newly collected examples of different styles, randomly split into 10 incremental tasks of disjoint classes. 5-Datasets [6] includes CIFAR-10 [14], MNIST [15], Fashion-MNIST [42], SVHN [24] and notMNIST [1] datasets, each treated as an incremental task to evaluate the impact of large inter-task differences. Split CUB-200 [32] includes 200-class fine-grained images of birds, randomly split into 10 incremental tasks of disjoint classes.

**Baseline:** We compare four representative prompt-based approaches as discussed in Sec. 3.1, such as L2P [41], DualPrompt [40], S-Prompt++ [39] and CODA-Prompt [30]. To evaluate the performance of continual learning, we record the average accuracy of all seen classes after learning each task, presenting the last one as the final average accuracy (FAA) and their historical average as the cumulative average accuracy (CAA) [37]. We also present the final forgetting measure (FFM) of all tasks [37]. We consider a variety of pre-training paradigms for ImageNet-21K and ImageNet-1K, including Sup-21K, iBOT-21K, iBOT-1K, DINO-1K and MoCo-1K, as described in Sec. 3.2.

**Implementation:** We follow similar implementations as previous work [41, 40, 30]. Specifically, we adopt a pre-trained ViT-B/16 backbone and train with an Adam optimizer ($\beta_1 = 0.9$, $\beta_2 = 0.999$), a batch size of 128, and a constant learning rate of 0.005 (except for CODA-Prompt with a cosine-decaying learning rate of 0.001), and grid search for a proper epoch number. The image inputs are resized to $224 \times 224$ and normalized to $[0, 1]$. Please refer to Appendix C for more details.

**Overall Performance:** Table 1 presents the main results of all approaches. Consistent with the observations in Sec. 3.2, representative prompt-based approaches achieve outstanding performance under supervised pre-training (i.e., Sup-21K), while perform significantly worse under the more realistic self-supervised pre-training. In particular, the most recent CODA-Prompt [30] outperforms other baselines in general (here we take FAA as the primary metric), but is sensitive to learning rate (see Fig. 2, a, b and Appendix Table 6). In contrast, our HiDe-Prompt achieves generally the

highest FAA, CAA and the lowest FFM, thus greatly superior to all competitors. This advantage is more pronounced under self-supervised pre-training, e.g., up to **15.01**% and **9.61**% lead on Split CIFAR-100 and Split ImageNet-R, respectively.[4] When considering large inter-task differences and fine-grained classification (see Table 2), the performance lead of our approach under self-supervised pre-training becomes even more significant, e.g., up to **12.63**% and **30.44**% on 5-Datasets and Split CUB-200, respectively. We further evaluate the impact of pre-trained knowledge that is disjoint from downstream tasks, following the setup of a recent work [12]. Specifically, we perform Split CIFAR-100 with pre-training on a subset of ImageNet in which 389 similar classes were removed. The FAAs of S-Prompt++, CODA-Prompt and HiDe-Prompt are 69.00%, 65.07% and 88.05%, respectively. All above results demonstrate the strength of our approach, thanks to the explicit optimization of the hierarchical components that overcomes potential sub-optimality.

Here we explore the characteristics of HiDe-Prompt in more depth. First, HiDe-Prompt requires much smaller GPU memory compared to CODA-Prompt (e.g., 16141MB vs 25325MB on Split CIFAR-100), since we optimize the construction and inference of appropriate prompts separately as hierarchical components rather than jointly in an end-to-end fashion. Compared to other baselines (except L2P that requires a much smaller epoch number), the computation cost of our approach is comparable in order of magnitude. Using the same single-card A100 GPU, for example, the training times of L2P, DualPrompt, S-Prompt++, CODA-Prompt and HiDe-Prompt are 0.55h, 2.00h, 2.01h, 2.08h, and 2.80h on Split CIFAR-100, respectively. As for the impact of statistical modeling, reserving around 5 centroids for each class performs comparably to a single Gaussian (see Appendix Table 7), which ensures storage efficiency and can potentially be generalized to other task types.

Table 2: Extended benchmarks. Here we present FAA (↑) for all approaches. The results of 5-Datasets are further detailed in Appendix Table 8.

| Method | 5-Datasets | | Split CUB-200 | |
|---|---|---|---|---|
| | Sup-21K | iBOT-21K | Sup-21K | iBOT-21K |
| L2P [41] | 81.84 | 82.25 | 74.48 | 44.29 |
| DualPrompt [40] | 77.91 | 68.03 | 82.05 | 41.31 |
| S-Prompt++ [39] | 86.06 | 77.20 | 82.08 | 42.73 |
| CODA-Prompt [30] | 64.18 | 51.65 | 74.34 | 47.79 |
| HiDe-Prompt (Ours) | **93.83** | **94.88** | **86.56** | **78.23** |

Table 3: Ablation study of hierarchical components. Here we present FAA (↑) for all baselines. "WTP" refers to the use of prompt ensemble within the naive architecture of task-specific prompts.

| Baseline | Split CIFAR-100 | | | | | Split ImageNet-R | | | | |
|---|---|---|---|---|---|---|---|---|---|---|
| | Sup-21K | iBOT-21K | iBOT-1K | DINO-1K | MoCo-1K | Sup-21K | iBOT-21K | iBOT-1K | DINO-1K | MoCo-1K |
| Naive Architecture | 85.11 | 73.05 | 72.20 | 73.74 | 75.67 | 60.22 | 48.00 | 53.68 | 54.33 | 48.77 |
| WTP | 87.86 | 78.86 | 75.93 | 75.15 | 77.15 | 71.57 | 55.16 | 60.86 | 57.61 | 53.21 |
| WTP+TII | 88.05 | 80.77 | 78.90 | 76.27 | 77.78 | 73.76 | 55.19 | 61.22 | 58.41 | 53.08 |
| WTP+TAP | 89.85 | 84.23 | 86.04 | 84.76 | 85.17 | 72.57 | 60.01 | 67.13 | 64.26 | 58.36 |
| WTP+TII+TAP | 92.50 | 90.21 | 90.52 | 88.93 | 89.28 | 74.89 | 70.44 | 70.66 | 66.78 | 63.59 |
| WTP+TII+TAP w/ CR | **92.61** | **93.02** | **93.48** | **92.51** | **91.57** | **75.06** | **70.83** | **71.33** | **68.11** | **63.77** |

Table 4: Effect of CR on WTP and the full HiDe-Prompt. Here we present FAA (↑) for all baselines.

| PTM | Baseline | Split CIFAR-100 | | | | Split ImageNet-R | | | |
|---|---|---|---|---|---|---|---|---|---|
| | | $\lambda = 0$ | 0.001 | 0.01 | 0.1 | $\lambda = 0$ | 0.001 | 0.01 | 0.1 |
| iBOT-1K | WTP w/ CR | **75.13** | 72.75 | 72.53 | 71.53 | **61.47** | 61.17 | 61.07 | 59.52 |
| | WTP+TII+TAP w/ CR | 90.15 | 91.18 | 91.60 | **93.35** | 70.97 | 71.25 | 71.57 | **71.71** |
| DINO-1K | WTP w/ CR | **74.99** | 73.23 | 72.74 | 70.87 | 57.71 | **58.23** | 57.35 | 56.37 |
| | WTP+TII+TAP w/ CR | 88.97 | 90.06 | 90.54 | **92.61** | 66.88 | 67.42 | 68.13 | **68.21** |

**Ablation Study:** We perform an extensive ablation study in Table 3 to validate the effectiveness of hierarchical components in HiDe-Prompt. We first construct a naive architecture of task-specific prompts as the baseline and then incorporate progressively the individual designs of Sec. 4.2. In general, the optimization of each component, such as within-task prediction (WTP), task-identity inference (TII) and task-adaptive prediction (TAP), delivers clear benefits and all contribute to the strong performance of HiDe-Prompt. Interestingly, the improvement of TII only becomes apparent

---

[4]A contemporaneous work [20] also reported FAA under DINO-1K on Split CIFAR-100, Split ImageNet-R and 5-Datasets (83.59%, 61.22% and 89.10%), which lags far behind ours (92.51%, 68.11% and 93.50%).

with WTP+TAP rather than with WTP only, suggesting that these hierarchical components are highly *synergistic* instead of operating in isolation. The proposed contrastive regularization (CR) helps the instructed representations of individual tasks to be compatible with each other and avoid inter-task overlap, thus further facilitating the performance of WTP+TII+TAP. Moreover, we observe that the improvements of WTP, TII, TAP and CR are generally more significant under *self-supervised pre-training*, due to effectively addressing the obscured sub-optimality.

**Detailed Analysis:** Now we further analyze the contributions of the three components for continual learning. First, we evaluate the average performance of learning each new task in Fig. 4, a, where WTP clearly outperforms the naive architecture, indicating that the knowledge of resolving each task is better incorporated into the task-specific prompts. Second, we present the accuracy of predicting task identity from uninstructed representations in Fig. 4, b, which has been substantially improved

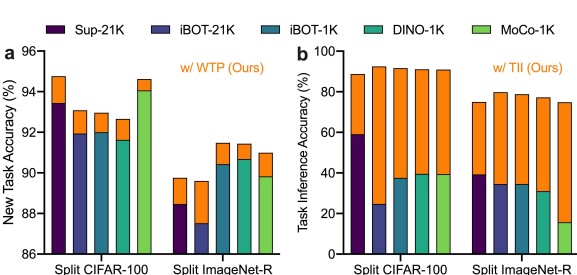

Figure 4: Detailed Analysis of WTP and TII.

(up to 67.74%) through optimizing TII explicitly. Third, we evaluate the effects of CR on only WTP and the full model of HiDe-Prompt (i.e., WTP+TII+TAP) in Table 4. Increasing the strength of CR decreases the performance of WTP since the task-specific prompt includes more knowledge about other tasks, but improves the performance of HiDe-Prompt since the inter-task representations become more compatible. These results suggest a potential *trade-off* between the knowledge for WTP and TAP, which can be modulated explicitly by CR.

# 6   Discussion and Conclusion

In this work, we analyze extensively the advanced prompt-based continual learning from both empirical and theoretical perspectives. An important finding is that, the continual learning objective (which can be decomposed into WTP, TII and TAP in the context of pre-training) is not adequately achieved, and this sub-optimality is clearly exposed under the more realistic self-supervised pre-training. By leveraging statistics of uninstructed and instructed representations, we present a strong approach to optimize explicitly the hierarchical components, which achieves superior performance across various pre-training paradigms. In particular, our theoretical analysis and the proposed approach can serve as a general framework for implementing parameter-efficient fine-tuning techniques (e.g., prompt, adapter, LoRA, FiLM, etc.) in continual learning[5], which differ only in the form of task-specific parameters. Interestingly, our proposal is consistent with recent advances in neuroscience [17, 16], where the activation of non-memory cells and memory cells (as well as their specific populations) is internally switched. Based on these results, we expect subsequent work to further explore the architecture and optimization of continual learning with an effective use of pre-trained knowledge.

This work remains some potential *limitations*. First, we assume adequate pre-training to provide meaningful representations, which may not be available in some applications. Second, the prompt-based strategy is mainly applicable to the transformer backbone rather than other backbone architectures. Third, the transformer backbone is frozen and adapted to downstream tasks via prompt parameters, which prevents the pre-trained knowledge from being enriched and updated. As a fundamental research in machine learning, the potential *negative societal impact* is not obvious at this stage.

# Acknowledgements

This work was supported by the National Key Research and Development Program of China (No. 2020AAA0106302), NSFC Projects (Nos. 62061136001, 92248303, 62106123, 61972224), BNRist (BNR2022RC01006), Tsinghua Institute for Guo Qiang, and the High Performance Computing Center, Tsinghua University. L.W. is also supported by Shuimu Tsinghua Scholar, and J.Z. is also supported by the XPlorer Prize.

---

[5]In fact, our LoRA version is also competitive, e.g., the FAA is 87.53% on Split CIFAR-100 under iBOT-21K.

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

**Algorithm 1** Training Algorithm of HiDe-Prompt

---

**Input**: Pre-trained transformer backbone $f_\theta$, training sets $\mathcal{D}_1, ..., \mathcal{D}_T$, number of tasks $T$, number of epochs $E$, hyperparameters $\alpha$, $\tau$ and $\lambda$.
**Output**: Parameters $\boldsymbol{p}_1, ..., \boldsymbol{p}_T$, $\omega$ and $\psi$

```
 1: Initialize e₁, ω and ψ
 2: for t = 1, ..., T do
 3:     for c ∈ 𝒴_t do
 4:         Obtain Ĝ_c from f_θ and 𝒟_t                          ▷ Uninstructed Representations
 5:     if t > 1 then
 6:         Initialize e_t ← e_{t−1}
 7:         Construct p_t = α ∑_{i=1}^{t−1} e_i + (1 − α)e_t
 8:     else
 9:         Construct p_t = e_t
10:     for epoch = 1, ..., E do
11:         Optimize p_t and ψ with ℒ_WTP in Eq. (10)             ▷ Within-Task Prediction
12:         Optimize ω with ℒ_TII in Eq. (11)                      ▷ Task-Identity Inference
13:         Optimize ψ with ℒ_TAP in Eq. (12)                      ▷ Task-Adaptive Prediction
14:     for c ∈ 𝒴_t do
15:         Obtain G_c from f_θ, p_t and 𝒟_t                      ▷ Instructed Representations
16: return (p_1, ..., p_T, ω, ψ)
```

---

# A  Theoretical Foundation

## A.1  Class-Incremental Learning (CIL)

**Proof of Theorem 1**

*Proof.* For class-incremental learning (CIL) with pre-training, assume $\mathbb{E}_{\boldsymbol{x}}[H_{\text{WTP}}(\boldsymbol{x})] \leq \delta$, $\mathbb{E}_{\boldsymbol{x}}[H_{\text{TII}}(\boldsymbol{x})] \leq \epsilon$, and $\mathbb{E}_{\boldsymbol{x}}[H_{\text{TAP}}(\boldsymbol{x})] \leq \eta$. Let $y = \mathcal{Y}_{\bar{i},\bar{j}}$ be the ground truth of an $\boldsymbol{x}$, where $\bar{i} \in \{1, ..., t\}$ and $\bar{j} \in \{1, ..., |\mathcal{Y}_i|\}$ denote the task identity and within-task index, respectively.

As we defined,

$$
\begin{aligned}
H_{\text{WTP}}(\boldsymbol{x}) &= \mathcal{H}(\mathbf{1}_{\bar{j}}, \{P(\boldsymbol{x} \in \mathcal{X}_{\bar{i},j} | \boldsymbol{x} \in \mathcal{X}_{\bar{i}}, \mathcal{D}, \theta)\}_j) \\
&= -\log P(\boldsymbol{x} \in \mathcal{X}_{\bar{i},\bar{j}} | \boldsymbol{x} \in \mathcal{X}_{\bar{i}}, \mathcal{D}, \theta),
\end{aligned}
\tag{13}
$$

$$
\begin{aligned}
H_{\text{TII}}(\boldsymbol{x}) &= \mathcal{H}(\mathbf{1}_{\bar{i}}, \{P(\boldsymbol{x} \in \mathcal{X}_i | \mathcal{D}, \theta)\}_i) \\
&= -\log P(\boldsymbol{x} \in \mathcal{X}_{\bar{i}} | \mathcal{D}, \theta),
\end{aligned}
\tag{14}
$$

$$
\begin{aligned}
H_{\text{TAP}}(\boldsymbol{x}) &= \mathcal{H}(\mathbf{1}_{\bar{c}}, \{P(\boldsymbol{x} \in \mathcal{X}^c | \mathcal{D}, \theta)\}_c) \\
&= -\log P(\boldsymbol{x} \in \mathcal{X}^{\bar{c}} | \mathcal{D}, \theta) \\
&= -\log P(\boldsymbol{x} \in \mathcal{X}^y | \mathcal{D}, \theta).
\end{aligned}
\tag{15}
$$

Then, we have

$$
\begin{aligned}
&\mathcal{H}(\mathbf{1}_{\bar{i},\bar{j}}, \{P(\boldsymbol{x} \in \mathcal{X}_{i,j} | \mathcal{D}, \theta)\}_{i,j}) \\
&= -\log P(\boldsymbol{x} \in \mathcal{X}_{\bar{i},\bar{j}} | \mathcal{D}, \theta) \\
&= -\log(P(\boldsymbol{x} \in \mathcal{X}_{\bar{i},\bar{j}} | \boldsymbol{x} \in \mathcal{X}_{\bar{i}}, \mathcal{D}, \theta)P(\boldsymbol{x} \in \mathcal{X}_{\bar{i}} | \mathcal{D}, \theta)) \\
&= -\log P(\boldsymbol{x} \in \mathcal{X}_{\bar{i},\bar{j}} | \boldsymbol{x} \in \mathcal{X}_{\bar{i}}, \mathcal{D}, \theta) - \log P(\boldsymbol{x} \in \mathcal{X}_{\bar{i}} | \mathcal{D}, \theta) \\
&= H_{\text{WTP}}(\boldsymbol{x}) + H_{\text{TII}}(\boldsymbol{x}).
\end{aligned}
\tag{16}
$$

Taking expectations on Eq. (15), we have

$$
\mathcal{L}_1 = \mathbb{E}_{\boldsymbol{x}}[H_{\text{TAP}}(\boldsymbol{x})] \leq \eta.
\tag{17}
$$

Taking expectations on both sides of Eq. (16), we have

$$
\begin{aligned}
\mathcal{L}_2 &= \mathbb{E}_{\boldsymbol{x}}[\mathcal{H}(\mathbf{1}_{\bar{i},\bar{j}}, \{P(\boldsymbol{x} \in \mathcal{X}_{i,j} | \mathcal{D}, \theta)\}_{i,j})] \\
&= \mathbb{E}_{\boldsymbol{x}}[H_{\text{WTP}}(\boldsymbol{x})] + \mathbb{E}_{\boldsymbol{x}}[H_{\text{TII}}(\boldsymbol{x})] \\
&\leq \delta + \epsilon.
\end{aligned}
\tag{18}
$$

Since our objective of CIL with pre-training is $\max[P(\boldsymbol{x} \in \mathcal{X}_{\bar{i},\bar{j}}|\mathcal{D}, \theta), P(\boldsymbol{x} \in \mathcal{X}^y|\mathcal{D}, \theta)]$, then we have the loss error

$$
\begin{aligned}
\mathcal{L} &= \max\{\mathbb{E}_{\boldsymbol{x}}[\mathcal{H}(\mathbf{1}_{\bar{i},\bar{j}}, \{P(\boldsymbol{x} \in \mathcal{X}_{i,j}|\mathcal{D}, \theta)\}_{i,j})], \mathbb{E}_{\boldsymbol{x}}[H_{\text{TAP}}(\boldsymbol{x})]\} \\
&= \max\{\mathcal{L}_2, \mathcal{L}_1\} \\
&= \max\{\delta + \epsilon, \eta\}.
\end{aligned}
\tag{19}
$$

This finishes the proof.

**Proof of Theorem 2**

*Proof.* For CIL with pre-training, its loss error $\mathcal{L} \leq \xi$. Assume $\boldsymbol{x} \in \mathcal{X}_{\bar{i},\bar{j}} \subseteq \mathcal{X}_{\bar{i}}$. According to the proof of Theorem 1, we have

$$
\begin{aligned}
H_{\text{WTP}}(\boldsymbol{x}) &= -\log P(\boldsymbol{x} \in \mathcal{X}_{\bar{i},\bar{j}}|\boldsymbol{x} \in \mathcal{X}_{\bar{i}}, \mathcal{D}, \theta) \\
&= -\log \frac{P(\boldsymbol{x} \in \mathcal{X}_{\bar{i},\bar{j}}|\mathcal{D}, \theta)}{P(\boldsymbol{x} \in \mathcal{X}_{\bar{i}}|\mathcal{D}, \theta)} \\
&\leq -\log P(\boldsymbol{x} \in \mathcal{X}_{\bar{i},\bar{j}}|\mathcal{D}, \theta) \\
&= \mathcal{H}(\mathbf{1}_{\bar{i},\bar{j}}, \{P(\boldsymbol{x} \in \mathcal{X}_{i,j}|\mathcal{D}, \theta)\}_{i,j}) \\
&= \mathcal{L}_2 \leq \xi.
\end{aligned}
\tag{20}
$$

Likewise, we have

$$
\begin{aligned}
H_{\text{TII}}(\boldsymbol{x}) &= -\log P(\boldsymbol{x} \in \mathcal{X}_{\bar{i}}|\mathcal{D}, \theta) \\
&= -\log \frac{P(\boldsymbol{x} \in \mathcal{X}_{\bar{i},\bar{j}}|\mathcal{D}, \theta)}{P(\boldsymbol{x} \in \mathcal{X}_{\bar{i},\bar{j}}|\boldsymbol{x} \in \mathcal{X}_{\bar{i}}, \mathcal{D}, \theta)} \\
&\leq -\log P(\boldsymbol{x} \in \mathcal{X}_{\bar{i},\bar{j}}|\mathcal{D}, \theta) \\
&= \mathcal{H}(\mathbf{1}_{\bar{i},\bar{j}}, \{P(\boldsymbol{x} \in \mathcal{X}_{i,j}|\mathcal{D}, \theta)\}_{i,j}) \\
&= \mathcal{L}_2 \leq \xi.
\end{aligned}
\tag{21}
$$

In addition, we have formulated the final goal of CIL as a multi-objective optimization problem, i.e., $\max[P(\boldsymbol{x} \in \mathcal{X}_{\bar{i},\bar{j}}|\mathcal{D}, \theta), P(\boldsymbol{x} \in \mathcal{X}^y|\mathcal{D}, \theta)]$. Then, each objective must guarantee the loss error less than $\xi$, i.e.,

$$
\begin{aligned}
H_{\text{TAP}}(\boldsymbol{x}) &= -\log P(\boldsymbol{x} \in \mathcal{X}^y|\mathcal{D}, \theta) \\
&= \mathcal{L}_1 \leq \xi.
\end{aligned}
\tag{22}
$$

This finishes the proof.

## A.2  Domain-Incremental Learning (DIL)

For domain-incremental learning (DIL), Let $\mathcal{X}_t = \bigcup_j \mathcal{X}_{t,j}$ and $\mathcal{Y}_t = \{\mathcal{Y}_{t,j}\}$, where $j \in \{1, ..., |\mathcal{Y}_t|\}$ denotes the $j$-th class in task $t$. Now assume we have a ground event denoted as $\mathcal{D} = \{\mathcal{D}_1, ..., \mathcal{D}_t\}$ and a pre-trained model $f_\theta$. For any sample $\boldsymbol{x} \in \bigcup_{k=1}^t \mathcal{X}_k$, a general goal of the DIL problem is to learn $P(\boldsymbol{x} \in \mathcal{X}_{*,j}|\mathcal{D}, \theta)$, where $\mathcal{X}_{*,j}$ represents the $j$-th class domain in any task. Of note, $\mathcal{Y}_t = \mathcal{Y}_{t'}$, $\forall t \neq t'$ for DIL. This can be decomposed into two probabilities, including task-identity inference (TII) and within-task prediction (WTP), denoted as $P(\boldsymbol{x} \in \mathcal{X}_i|\mathcal{D}, \theta)$ and $P(\boldsymbol{x} \in \mathcal{X}_{i,j}|\boldsymbol{x} \in \mathcal{X}_i, \mathcal{D}, \theta)$, respectively. Based on Bayes' theorem, we have

$$
P(\boldsymbol{x} \in \mathcal{X}_{*,j}|\mathcal{D}, \theta) = \sum_i P(\boldsymbol{x} \in \mathcal{X}_{i,j}|\boldsymbol{x} \in \mathcal{X}_i, \mathcal{D}, \theta)P(\boldsymbol{x} \in \mathcal{X}_i|\mathcal{D}, \theta),
\tag{23}
$$

where $\{*, j\}$ represents the $j$-th class in each domain.

Then we have the following theorems in terms of the sufficient and necessary conditions for improving DIL with pre-training.

**Theorem 3** *For domain-incremental learning with pre-training, if $\mathbb{E}_{\boldsymbol{x}}[H_{\text{WTP}}(\boldsymbol{x})] \leq \delta$, $\mathbb{E}_{\boldsymbol{x}}[H_{\text{TII}}(\boldsymbol{x})] \leq \epsilon$, and $\mathbb{E}_{\boldsymbol{x}}[H_{\text{TAP}}(\boldsymbol{x})] \leq \eta$, we have the loss error $\mathcal{L} \in [0, \max\{\delta + \epsilon + \log t, \eta\}]$, regardless whether WTP, TII and TAP are trained together or separately.*

**Proof of Theorem 3**

*Proof.* Let $\bar{j} \in \{1, ..., |\mathcal{Y}_t|\}$ and $y \in \mathcal{Y}_t$ be the ground truth of an $\boldsymbol{x}$ w.r.t. the within-task index and class label, and $y = \mathcal{Y}_{i,\bar{j}}$ for any $i \in \{1, ..., t\}$. Eq. (23) suggests that if we can improve either the WTP performance $P(\boldsymbol{x} \in \mathcal{X}_{i,\bar{j}} | \boldsymbol{x} \in \mathcal{X}_i, \mathcal{D}, \theta)$, the TII performance $P(\boldsymbol{x} \in \mathcal{X}_i | \mathcal{D}, \theta)$, or both, then the DIL performance $P(\boldsymbol{x} \in \mathcal{X}^y | \mathcal{D}, \theta)$ would be improved. However, such an improvement is limited since it is upper-bounded by WTP or TII. To further improve the DIL performance, we propose a hierarchical decomposition of the objective. That is, besides the improvement of $P(\boldsymbol{x} \in \mathcal{X}_{*,\bar{j}} | \mathcal{D}, \theta)$, we also need to directly improve the performance of task-adaptive prediction (TAP), denoted as $P(\boldsymbol{x} \in \mathcal{X}^y | \mathcal{D}, \theta)$, where $y \in \{1, ..., |\mathcal{Y}_t|\}$, $\mathcal{X}^y$ represents the domain of class $y$ in all observed domains, and $\mathcal{X}^y = \bigcup_i \mathcal{X}_{i,\bar{j}}$. Then the final goal of DIL is formulated as a multi-objective optimization problem, i.e., $\max[P(\boldsymbol{x} \in \mathcal{X}_{*,\bar{j}} | \mathcal{D}, \theta), P(\boldsymbol{x} \in \mathcal{X}^y | \mathcal{D}, \theta)]$. Notice that the TII probability is a categorical distribution over all observed domains $\{1 : t\}$, while the TAP probability is over all observed classes $\bigcup_{k=1}^t \mathcal{Y}_k$.

As similarly defined in CIL, here

$$
\begin{aligned}
H_{\text{WTP}}(\boldsymbol{x}) &= \mathcal{H}(\mathbf{1}_{\bar{j}}, \{P(\boldsymbol{x} \in \mathcal{X}_{i,j} | \boldsymbol{x} \in \mathcal{X}_i, \mathcal{D}, \theta)\}_j) \\
&= -\log P(\boldsymbol{x} \in \mathcal{X}_{i,\bar{j}} | \boldsymbol{x} \in \mathcal{X}_i, \mathcal{D}, \theta),
\end{aligned}
\tag{24}
$$

$$
\begin{aligned}
H_{\text{TII}}(\boldsymbol{x}) &= \mathcal{H}(\gamma, \{P(\boldsymbol{x} \in \mathcal{X}_i | \mathcal{D}, \theta)\}_i) \\
&= -\gamma_i \log P(\boldsymbol{x} \in \mathcal{X}_i | \mathcal{D}, \theta),
\end{aligned}
\tag{25}
$$

$$
\begin{aligned}
H_{\text{TAP}}(\boldsymbol{x}) &= \mathcal{H}(\mathbf{1}_{\bar{c}}, \{P(\boldsymbol{x} \in \mathcal{X}^c | \mathcal{D}, \theta)\}_c) \\
&= -\log P(\boldsymbol{x} \in \mathcal{X}^{\bar{c}} | \mathcal{D}, \theta) \\
&= -\log P(\boldsymbol{x} \in \mathcal{X}^y | \mathcal{D}, \theta),
\end{aligned}
\tag{26}
$$

where $\gamma = \{\gamma_i\}_{i=1}^t$ represents the possibility of $\boldsymbol{x}$ belonging to each observed domain, $\gamma_i \in [0, 1]$ and $\sum_i \gamma_i = 1$.

Then, for any simplex $\gamma$, we have

$$
\begin{aligned}
&\mathcal{H}(\mathbf{1}_{\bar{j}}, \{P(\boldsymbol{x} \in \mathcal{X}_{*,j} | \mathcal{D}, \theta)\}_j) \\
&= -\log P(\boldsymbol{x} \in \mathcal{X}_{*,\bar{j}} | \mathcal{D}, \theta) \\
&= -\log(\sum_i P(\boldsymbol{x} \in \mathcal{X}_{i,\bar{j}} | \boldsymbol{x} \in \mathcal{X}_i, \mathcal{D}, \theta) P(\boldsymbol{x} \in \mathcal{X}_i | \mathcal{D}, \theta)) \\
&\leq -\sum_i \gamma_i \log(\frac{P(\boldsymbol{x} \in \mathcal{X}_{i,\bar{j}} | \boldsymbol{x} \in \mathcal{X}_i, \mathcal{D}, \theta) P(\boldsymbol{x} \in \mathcal{X}_i | \mathcal{D}, \theta)}{\gamma_i}) \\
&= -\sum_i \gamma_i \log P(\boldsymbol{x} \in \mathcal{X}_{i,\bar{j}} | \boldsymbol{x} \in \mathcal{X}_i, \mathcal{D}, \theta) - \sum_i \gamma_i \log P(\boldsymbol{x} \in \mathcal{X}_i | \mathcal{D}, \theta) + \sum_i \gamma_i \log(\gamma_i) \\
&= \sum_i \gamma_i H_{\text{WTP}} + H_{\text{TII}} + \mathcal{H}(\gamma).
\end{aligned}
\tag{27}
$$

Taking expectations on Eq. (26), we have

$$
\mathcal{L}_1 = \mathbb{E}_{\boldsymbol{x}}[H_{\text{TAP}}(\boldsymbol{x})] \leq \eta.
\tag{28}
$$

Taking expectations on both sides of Eq. (27), we have

$$
\begin{aligned}
\mathcal{L}_2 &= \mathbb{E}_{\boldsymbol{x}}[\mathcal{H}(\mathbf{1}_{\bar{j}}, \{P(\boldsymbol{x} \in \mathcal{X}_{*,j} | \mathcal{D}, \theta)\}_j] \\
&\leq \sum_i \gamma_i \mathbb{E}_{\boldsymbol{x}}[H_{\text{WTP}}(\boldsymbol{x})] + \mathbb{E}_{\boldsymbol{x}}[H_{\text{TII}}(\boldsymbol{x})] + \mathcal{H}(\gamma) \\
&\leq \delta + \epsilon + \log t.
\end{aligned}
\tag{29}
$$

Since our objective of DIL with pre-training is $\max[P(\boldsymbol{x} \in \mathcal{X}_{*,\bar{j}} | \mathcal{D}, \theta), P(\boldsymbol{x} \in \mathcal{X}^y | \mathcal{D}, \theta)]$, then we have the loss error

$$
\begin{aligned}
\mathcal{L} &= \max\{\mathbb{E}_{\boldsymbol{x}}[\mathcal{H}(\mathbf{1}_{\bar{j}}, \{P(\boldsymbol{x} \in \mathcal{X}_{*,j} | \mathcal{D}, \theta)\}_j)], \mathbb{E}_{\boldsymbol{x}}[H_{\text{TAP}}(\boldsymbol{x})]\} \\
&= \max\{\mathcal{L}_2, \mathcal{L}_1\} \\
&= \max\{\delta + \epsilon + \log t, \eta\}.
\end{aligned}
\tag{30}
$$

This finishes the proof.

**Theorem 4** *For domain-incremental learning with pre-training, if the loss error $\mathcal{L} \leq \xi$, then there always exist (1) a WTP, s.t. $H_{\mathrm{WTP}} \leq \xi$; (2) a TII, s.t. $H_{\mathrm{TII}} \leq \xi$; and (3) a TAP, s.t. $H_{\mathrm{TAP}} \leq \xi$.*

**Proof of Theorem 4** For DIL with pre-training, its loss error $\mathcal{L} = \max[\mathcal{L}_1, \mathcal{L}_2] \leq \xi$. Assume $\boldsymbol{x} \in \mathcal{X}_{*,\bar{j}} \subseteq \mathcal{X}^y$. According to the proof of Theorem 3, if we define $P(\boldsymbol{x} \in \mathcal{X}_{i,\bar{j}}|\mathcal{D},\theta) = P(\boldsymbol{x} \in \mathcal{X}_{*,\bar{j}}|\mathcal{D},\theta)$, we have

$$
\begin{aligned}
H_{\mathrm{WTP}}(\boldsymbol{x}) &= -\log P(\boldsymbol{x} \in \mathcal{X}_{i,\bar{j}}|\boldsymbol{x} \in \mathcal{X}_i, \mathcal{D}, \theta) \\
&= -\log \frac{P(\boldsymbol{x} \in \mathcal{X}_{i,\bar{j}}|\mathcal{D},\theta)}{P(\boldsymbol{x} \in \mathcal{X}_i|\mathcal{D},\theta)} \\
&\leq -\log P(\boldsymbol{x} \in \mathcal{X}_{i,\bar{j}}|\mathcal{D},\theta) \\
&= -\log P(\boldsymbol{x} \in \mathcal{X}_{*,\bar{j}}|\mathcal{D},\theta) \\
&= \mathcal{H}(\mathbf{1}_{\bar{j}}, \{P(\boldsymbol{x} \in \mathcal{X}_{*,j}|\mathcal{D},\theta)\}_j) \\
&= \mathcal{L}_2 \leq \xi.
\end{aligned}
\tag{31}
$$

Likewise, if we define $\gamma_i = 1$ and $\gamma_{i'} = 0$ for $i' \neq i$, we have

$$
\begin{aligned}
H_{\mathrm{TII}}(\boldsymbol{x}) &= -\sum_i \gamma_i \log P(\boldsymbol{x} \in \mathcal{X}_i|\mathcal{D},\theta) \\
&= -\log P(\boldsymbol{x} \in \mathcal{X}_i|\mathcal{D},\theta) \\
&= -\log \frac{P(\boldsymbol{x} \in \mathcal{X}_{i,\bar{j}}|\mathcal{D},\theta)}{P(\boldsymbol{x} \in \mathcal{X}_{i,\bar{j}}|\boldsymbol{x} \in \mathcal{X}_i, \mathcal{D}, \theta)} \\
&\leq -\log(\boldsymbol{x} \in \mathcal{X}_{i,\bar{j}}|\mathcal{D},\theta) \\
&= -\log(\boldsymbol{x} \in \mathcal{X}_{*,\bar{j}}|\mathcal{D},\theta) \\
&= \mathcal{H}(\mathbf{1}_{\bar{j}}, \{P(\boldsymbol{x} \in \mathcal{X}_{*,j}|\mathcal{D},\theta)\}_j) \\
&= \mathcal{L}_2 \leq \xi.
\end{aligned}
\tag{32}
$$

In addition, we have formulated the final goal of DIL as a multi-objective optimization problem, i.e., $\max[P(\boldsymbol{x} \in \mathcal{X}_{*,\bar{j}}|\mathcal{D},\theta), P(\boldsymbol{x} \in \mathcal{X}^y|\mathcal{D},\theta)]$. Then, each objective must guarantee the loss error less than $\xi$, i.e.,

$$
\begin{aligned}
H_{\mathrm{TAP}}(\boldsymbol{x}) &= -\log P(\boldsymbol{x} \in \mathcal{X}^y|\mathcal{D},\theta) \\
&= \mathcal{L}_1 \leq \xi.
\end{aligned}
\tag{33}
$$

This finishes the proof.

### A.3 Task-Incremental Learning (TIL)

For task-incremental learning (TIL), let $\mathcal{X}_t = \bigcup_j \mathcal{X}_{t,j}$ and $\mathcal{Y}_t = \{\mathcal{Y}_{t,j}\}$, where $j \in \{1, ..., |\mathcal{Y}_t|\}$ indicates the $j$-th class in task $t$. Now assume we have a ground event denoted as $\mathcal{D} = \{\mathcal{D}_1, ..., \mathcal{D}_t\}$ and a pre-trained model $f_\theta$. For any sample $\boldsymbol{x} \in \bigcup_{k=1}^t \mathcal{X}_k$, a general goal of the TIL problem is to learn $P(\boldsymbol{x} \in \mathcal{X}_{\bar{i},j}|\boldsymbol{x} \in \mathcal{X}_{\bar{i}}, \mathcal{D}, \theta)$, where $\bar{i} \in \{1, ..., t\}$ and $j \in \{1, ..., |\mathcal{Y}_{\bar{i}}|\}$. This can be equivalent to within-task prediction (WTP). Different from CIL, the task identity is provided in TIL. Unlike DIL, $\mathcal{Y}_t \cap \mathcal{Y}_{t'} = \emptyset, \forall t \neq t'$. Then we have the following theorems in terms of the sufficient and necessary conditions for improving TIL with pre-training.

**Theorem 5** *For task-incremental learning with pre-training, $\mathbb{E}_{\boldsymbol{x}}[H_{\mathrm{TII}}(\boldsymbol{x})] = 0$, and task-adaptive prediction (TAP) is degraded into within-task prediction (WTP). If $\mathbb{E}_{\boldsymbol{x}}[H_{\mathrm{WTP}}(\boldsymbol{x})] \leq \delta$, we have the loss error $\mathcal{L} \in [0, \delta]$.*

**Proof of Theorem 5**

*Proof.* For task-incremental learning (TIL) with pre-training, assume $\mathbb{E}_{\boldsymbol{x}}[H_{\mathrm{WTP}}(\boldsymbol{x})] \leq \delta$. Given an $\boldsymbol{x}$ with the task identity $\bar{i} \in \{1, ..., t\}$, let $\bar{j} \in \{1, ..., |\mathcal{Y}_{\bar{i}}|\}$ be the ground truth of $\boldsymbol{x}$ w.r.t. the within-task index, and $y = \mathcal{Y}_{\bar{i},\bar{j}}$ be the ground truth label of $\boldsymbol{x}$.

As similarly defined in CIL, here

$$
\begin{aligned}
H_{\mathrm{WTP}}(\boldsymbol{x}) &= \mathcal{H}(\mathbf{1}_{\bar{j}}, \{P(\boldsymbol{x} \in \mathcal{X}_{\bar{i},j}|\boldsymbol{x} \in \mathcal{X}_{\bar{i}}, \mathcal{D}, \theta)\}_j) \\
&= -\log P(\boldsymbol{x} \in \mathcal{X}_{\bar{i},\bar{j}}|\boldsymbol{x} \in \mathcal{X}_{\bar{i}}, \mathcal{D}, \theta),
\end{aligned}
\tag{34}
$$

$$H_{\text{TII}}(\boldsymbol{x}) = \mathcal{H}(\mathbf{1}_{\bar{i}}, \{P(\boldsymbol{x} \in \mathcal{X}_i | \mathcal{D}, \theta)\}_i)$$
$$= -\log P(\boldsymbol{x} \in \mathcal{X}_{\bar{i}} | \mathcal{D}, \theta) \tag{35}$$
$$= -\log 1 = 0,$$

$$H_{\text{TAP}}(\boldsymbol{x}) = \mathcal{H}(\mathbf{1}_{\bar{c}}, \{P(\boldsymbol{x} \in \mathcal{X}^c | \boldsymbol{x} \in \mathcal{X}_{\bar{i}}, \mathcal{D}, \theta)\}_c)$$
$$= -\log P(\boldsymbol{x} \in \mathcal{X}^{\bar{c}} | \boldsymbol{x} \in \mathcal{X}_{\bar{i}}, \mathcal{D}, \theta)$$
$$= -\log P(\boldsymbol{x} \in \mathcal{X}^y | \boldsymbol{x} \in \mathcal{X}_{\bar{i}}, \mathcal{D}, \theta) \tag{36}$$
$$= H_{\text{WTP}}(\boldsymbol{x}).$$

Then, we have

$$\mathcal{H}(\mathbf{1}_{\bar{i},\bar{j}}, \{P(\boldsymbol{x} \in \mathcal{X}_{i,j} | \mathcal{D}, \theta)\}_{i,j})$$
$$= -\log P(\boldsymbol{x} \in \mathcal{X}_{\bar{i},\bar{j}} | \mathcal{D}, \theta)$$
$$= -\log(P(\boldsymbol{x} \in \mathcal{X}_{\bar{i},\bar{j}} | \boldsymbol{x} \in \mathcal{X}_{\bar{i}}, \mathcal{D}, \theta) P(\boldsymbol{x} \in \mathcal{X}_{\bar{i}} | \mathcal{D}, \theta))$$
$$= -\log P(\boldsymbol{x} \in \mathcal{X}_{\bar{i},\bar{j}} | \boldsymbol{x} \in \mathcal{X}_{\bar{i}}, \mathcal{D}, \theta) - \log P(\boldsymbol{x} \in \mathcal{X}_{\bar{i}} | \mathcal{D}, \theta) \tag{37}$$
$$= H_{\text{WTP}}(\boldsymbol{x}) + H_{\text{TII}}(\boldsymbol{x})$$
$$= H_{\text{WTP}}(\boldsymbol{x}).$$

Taking expectations on both sides of Eq. (37), we have

$$\mathcal{L} = \mathbb{E}_{\boldsymbol{x}}[\mathcal{H}(\mathbf{1}_{\bar{i},\bar{j}}, \{P(\boldsymbol{x} \in \mathcal{X}_{i,j} | \mathcal{D}, \theta)\}_{i,j})]$$
$$= \mathbb{E}_{\boldsymbol{x}}[H_{\text{WTP}}(\boldsymbol{x})] \tag{38}$$
$$\leq \delta.$$

Since our objective of TIL with pre-training is $P(\boldsymbol{x} \in \mathcal{X}_{\bar{i},\bar{j}} | \boldsymbol{x} \in \mathcal{X}_{\bar{i}}, \mathcal{D}, \theta)$, then we have the loss error $\mathcal{L} \leq \delta$. This finishes the proof.

**Theorem 6** *For task-incremental learning with pre-training, if the loss error $\mathcal{L} \leq \xi$, then there always exists a WTP, s.t. $H_{\text{WTP}} \leq \xi$.*

### Proof of Theorem 6

For TIL with pre-training, its loss error $\mathcal{L} \leq \xi$. Assume $\boldsymbol{x} \in \mathcal{X}_{\bar{i},\bar{j}} \subseteq \mathcal{X}_{\bar{i}}$. According to the proof of Theorem 5, we have

$$H_{\text{WTP}}(\boldsymbol{x}) = -\log P(\boldsymbol{x} \in \mathcal{X}_{\bar{i},\bar{j}} | \boldsymbol{x} \in \mathcal{X}_{\bar{i}}, \mathcal{D}, \theta)$$
$$= -\log \frac{P(\boldsymbol{x} \in \mathcal{X}_{\bar{i},\bar{j}} | \mathcal{D}, \theta)}{P(\boldsymbol{x} \in \mathcal{X}_{\bar{i}} | \mathcal{D}, \theta)}$$
$$\leq -\log P(\boldsymbol{x} \in \mathcal{X}_{\bar{i},\bar{j}} | \mathcal{D}, \theta) \tag{39}$$
$$= \mathcal{H}(\mathbf{1}_{\bar{i},\bar{j}}, \{P(\boldsymbol{x} \in \mathcal{X}_{i,j} | \mathcal{D}, \theta)\}_{i,j})$$
$$\leq \xi.$$

This finishes the proof.

## B  Impact of Pre-Training on Continual Learning

In this work, we focus on continual learning with pre-training, especially prompt-based continual learning that receives significant attention in this direction. Our theoretical contribution lies in the context of *pre-training*, where we demonstrate the sufficient and necessary conditions to achieve good continual learning performance. This is clearly different from those previous theorems on continual learning from scratch [13], which is analyzed below.

First, the condition is different. We formulate the hierarchical components as $\theta$-conditional probabilities, i.e., $P(\boldsymbol{x} \in \mathcal{X}_{\bar{i},j} | \boldsymbol{x} \in \mathcal{X}_{\bar{i}}, \mathcal{D}, \theta)$, $P(\boldsymbol{x} \in \mathcal{X}_i | \mathcal{D}, \theta)$ and $P(\boldsymbol{x} \in \mathcal{X}^c | \mathcal{D}, \theta)$ for WTP, TII and TAP, respectively, where $\theta$ captures the pre-trained knowledge in initialization. When training from scratch, since the randomly-initialized parameter set $\theta_0$ carries no information and is greatly different from the optimal solution, it needs be substantially changed in continual learning and should not be took

into account in objectives. In contrast, the pre-trained parameter set $\theta$ carries beneficial knowledge for downstream tasks. If the pre-training is adequately strong, $\theta$ is already close to the optimal solution and only requires appropriate fine-tuning. Therefore, $\theta$ needs to be stabilized (usually frozen) in continual learning and should be considered in objectives.

Then, due to the incorporation of $\theta$, the sufficient and necessary conditions to achieve good continual learning performance become different. In addition to WTP and TII, TAP is especially required in the problem of continual learning with pre-training. Without considering the impact of pre-training, i.e., training from scratch, TAP is equivalent to TII * WTP, i.e., $P(\boldsymbol{x} \in \mathcal{X}^y | \mathcal{D}, \theta) = P(\boldsymbol{x} \in \mathcal{X}_{\bar{i}} | \mathcal{D}, \theta) P(\boldsymbol{x} \in \mathcal{X}_{\bar{i}, \bar{j}} | \boldsymbol{x} \in \mathcal{X}_{\bar{i}}, \mathcal{D}, \theta)$, corresponding to $\delta + \epsilon = \eta$ in our Theorem 1. Therefore, TAP is not required when training from scratch, consistent with the theorems in [13]. On the other hand, with considering the impact of pre-training, TAP and TII * WTP are formulated as two different prediction problems where $P(\boldsymbol{x} \in \mathcal{X}^y | \mathcal{D}, \theta) \neq P(\boldsymbol{x} \in \mathcal{X}_{\bar{i}} | \mathcal{D}, \theta) P(\boldsymbol{x} \in \mathcal{X}_{\bar{i}, \bar{j}} | \boldsymbol{x} \in \mathcal{X}_{\bar{i}}, \mathcal{D}, \theta)$. Therefore, the final performance of having TAP in continual learning can outperform that without TAP, i.e., $\max[P(\boldsymbol{x} \in \mathcal{X}_{\bar{i}, \bar{j}} | \mathcal{D}, \theta), P(\boldsymbol{x} \in \mathcal{X}^y | \mathcal{D}, \theta)] > P(\boldsymbol{x} \in \mathcal{X}_{\bar{i}, \bar{j}} | \mathcal{D}, \theta)$.

In the case of prompt-based continual learning, without considering the impact of pre-training, TAP = TII * WTP, due to classifying the same input in the same semantic space, i.e., $\frac{\exp(h_\psi(f(\boldsymbol{x}))[y])}{\sum_{j=1}^{t} \sum_{y' \in \mathcal{Y}_j} \exp(h_\psi(f(\boldsymbol{x}))[y'])} = \frac{\exp(\hat{h}_\omega(f(\boldsymbol{x})[\bar{i}])}{\sum_{i=1}^{t} \exp(\hat{h}_\omega(f(\boldsymbol{x})[i])} * \frac{\exp(h_\psi(f(\boldsymbol{x}))[\bar{j}])}{\sum_{j \in \mathcal{Y}_i} \exp(h_\psi(f(\boldsymbol{x}))[j])}$. With considering the impact of pre-training, TAP != TII * WTP, because TII * WTP is to calculate $\frac{\exp(\hat{h}_\omega(f(\boldsymbol{x})[\bar{i}])}{\sum_{i=1}^{t} \exp(\hat{h}_\omega(f(\boldsymbol{x})[i])}$ $* \frac{\exp(h_\psi(f(\boldsymbol{x}; \boldsymbol{p}_i))[\bar{j}])}{\sum_{j \in \mathcal{Y}_i} \exp(h_\psi(f(\boldsymbol{x}; \boldsymbol{p}_i))[j])}$, while TAP is to calculate $\frac{\exp(h_\psi(f(\boldsymbol{x}; \boldsymbol{p}_i))[y])}{\sum_{j=1}^{t} \sum_{y' \in \mathcal{Y}_j} \exp(h_\psi(f(\boldsymbol{x}; \boldsymbol{p}_i))[y'])}$. In other words, the backbone parameters are frozen to stabilize the pre-trained knowledge, with additional prompts $\boldsymbol{p}_i$ to fine-tune the semantic space (i.e., representations). TAP is essentially classifying the instructed representations, i.e., $\boldsymbol{h} \in \mathcal{H}_{i,c}$ in Eq. (12), while TII * WTP is performed on uninstructed representations, i.e., $\hat{h} \in \hat{\mathcal{H}}_{i,c}$ in Eq. (11). Targeting on different semantic spaces, TAP and TII * WTP are clearly different, i.e., TAP != TII * WTP. Strong empirical results also demonstrate their respective contributions to continual learning performance (see ablation study in Table 3).

## C   Implementation Details

In this section, we describe the implementation details of all experiments.

**Prompt-Based Approaches**: We follow the same implementations of the prompt architectures for all baselines as their original papers [41, 40, 30] (except S-Prompt++), which have been shown to yield strong performance. Specifically, L2P [41] is implemented with $M = 30$ as the total number of prompts ($M = 10$ [41] and $M = 30$ [40] have similar results), $L_{\boldsymbol{p}} = 5$ as the prompt length, and $N = 5$ for the Top-$N$ keys. DualPrompt [40] is implemented with $L_{\boldsymbol{g}} = 5$ as the prompt length of task-sharing prompts $\boldsymbol{g}$ inserted into layers 1-2 and $L_{\boldsymbol{e}} = 20$ as the prompt length of task-specific prompts $\boldsymbol{e}$ inserted into layers 3-5. S-Prompt++ [39] is implemented similarly to DualPrompt but replaces all task-sharing prompts with task-specific prompts, i.e., the task-specific prompts are inserted into layers 1-5 with prompt length $L_{\boldsymbol{e}} = 20$. CODA-Prompt [30] is implemented with $M = 100$ as the total number of prompts and $L_{\boldsymbol{p}} = 8$ as the prompt length, inserted into the same layers 1-5 as DualPrompt and S-Prompt++. HiDe-Prompt adopts a similar architecture as S-Prompt++, but replaces the task-specific keys with an auxiliary output layer $\hat{h}_\omega$ to predict the task identity and further preserves statistics of uninstructed and instructed representations. The hyperparameters for HiDe-Prompt are set to $\alpha = 0.1$, $\tau = 0.8$, and $\lambda = 0.1$.

**Training Regime**: Following the implementations of previous work [41, 40], we employ a pre-trained ViT-B/16 backbone, an Adam optimizer ($\beta_1 = 0.9$, $\beta_2 = 0.999$) and a batch size of 128. The learning rate is set to 0.001 with cosine decay for CODA-Prompt (empirically validated in Table 6), compared to 0.005 for other approaches. We then grid search for an appropriate number of epochs ($E$) under different pre-training paradigms, and observe that the best choice is generally consistent on each benchmark (i.e., $E$ is relatively insensitive to the pre-training paradigms). For Split CIFAR-100 with $E \in \{5, 10, 20, 40\}$, we set $E = 10$ for L2P and $E = 20$ for other approaches. For Split ImageNet-R with $E \in \{25, 50, 75, 150\}$, we set $E = 50$ for all approaches. For 5-Datasets with $E \in \{5, 10, 20, 40\}$, we set $E = 40$ for supervised pre-training and $E = 20$ for self-supervised pre-training. For Split CUB-200, we set $E = 50$ for all approaches.

**Evaluation Metric**: We focus on three evaluation metrics for continual learning, including the final average accuracy (FAA), cumulative average accuracy (CAA) and final forgetting measure (FFM). Specifically, we define the accuracy on the $i$-th task after learning the $t$-th task as $A_{i,t}$, and define the average accuracy of all seen tasks as $AA_t = \frac{1}{t} \sum_{i=1}^{t} A_{i,t}$. After learning all $T$ tasks, we report FAA $= AA_T$, CAA $= \frac{1}{T} \sum_{t=1}^{T} AA_t$, and FFM $= \frac{1}{T-1} \sum_{i=1}^{T-1} \max_{t \in \{1,...,T-1\}} (A_{i,t} - A_{i,T})$. FAA is the primary metric to evaluate the final performance of continual learning, CAA further reflects the historical performance, and FFM serves as a measure of catastrophic forgetting.

**Compute**: We run all experiments of Split CIFAR-100 on eight Tesla P100-SXM2 GPUs, Split ImageNet-R on four NVIDIA A100 GPUs, 5-Datasets on both, and Split CUB-200 on eight NVIDIA GeForce RTX 3090 GPUs.

# D   Extended Results

In this section, we provide some extended results for the main text.

**Reproduction of Baselines**: All results of L2P, DualPrompt and CODA-Prompt are reproduced from their official implementations, while S-Prompt++ is implemented with the code of DualPrompt. We compare the reported and reproduced results in Table 5. In particular, we use the same Sup-21K checkpoint as the original papers of L2P [41] and DualPrompt [40], and their reproduced results are comparable to the reported ones. In contrast, the original paper of CODA-Prompt [30] used a different supervised checkpoint, which is first pre-trained on ImageNet-21K in a self-supervised fashion and then fine-tuned on ImageNet-1K in a supervised fashion. Using the same supervised checkpoint as [30], we have justified that the results of CODA-Prompt on Split CIFAR-100 can be faithfully reproduced. As for Split ImageNet-R, we obtain the official code directly from the authors of CODA-Prompt, but it cannot reproduce exactly the reported results on Split ImageNet-R due to some clean-up issues. Consequently, the reproduced results of CODA-Prompt on Split ImageNet-R slightly underperform the reported ones under the same setting [30]. Besides, we observe that [30] used a different data split than originally proposed [40], resulting in an increase of around 3% in FAA. Together with the differences in supervised checkpoints, the reproduced results of CODA-Prompt in Table 1 and Table 5 differ by around 4% in FAA.

Table 5: Comparison of reported and reproduced results. *The forgetting metric for reported results [30] is implemented differently from ours.

| Baseline | Split CIFAR-100 | | Split ImageNet-R | |
|---|---|---|---|---|
| | FAA ($\uparrow$) | FFM ($\downarrow$) | FAA ($\uparrow$) | FFM ($\downarrow$) |
| L2P (Reported) [41] | 83.86 | 7.35 | 61.57 | 9.73 |
| L2P (Reproduced) | 83.06 | 6.58 | 63.65 | 7.51 |
| DualPrompt (Reported) [40] | 86.51 | 5.16 | 68.13 | 4.68 |
| DualPrompt (Reproduced) | 86.60 | 4.45 | 68.79 | 4.49 |
| CODA-Prompt (Reported) [30]* | 85.61 | 1.82 | 76.66 | 1.60 |
| CODA-Prompt (Reproduced) | 86.46 | 6.67 | 74.05 | 6.48 |

**Learning Rate of CODA-Prompt**: As discussed in the original paper of CODA-Prompt [30], its performance depends heavily on the reduced learning rates, especially on Split ImageNet-R. Here we extensively evaluate the effect of learning rate on the performance of CODA-Prompt (see Table 6), in order to justify the strength of our reproduced results under different pre-training paradigms. It can be clearly seen that using a smaller learning rate with cosine decay (i.e., LR=0.001 Cosine) is generally a good choice for CODA-Prompt, which is consistent with its original paper and therefore employed to reproduce its performance in this work.

**Statistical Modeling**: Here we evaluate the effect of statistical modeling. As analyzed in Sec. 4.2, preserving a dedicated mean and covariance for each class of representations can faithfully recover their distributions and thus achieves strong continual learning performance (see Table 7). Under Sup-21K that is adequately strong for downstream tasks, the performance remains essentially consistent when reducing the covariance to variance (e.g., the FAAs are 91.74% and 73.97% on Split CIFAR-100 and Split ImageNet-R, respectively). As a more generalized form of approximated distributions,

Table 6: Effect of learning rate on the performance of CODA-Prompt. Here we present FAA ($\uparrow$) for all baselines. $^*$The choice in its original paper [30].

| Learning Rate (LR) | Split CIFAR-100 | | | | | Split ImageNet-R | | | | |
|---|---|---|---|---|---|---|---|---|---|---|
| | Sup-21K | iBOT-21K | iBOT-1K | DINO-1K | MoCo-1K | Sup-21K | iBOT-21K | iBOT-1K | DINO-1K | MoCo-1K |
| LR=0.005 Constant | 85.92 | 76.46 | 72.39 | 71.43 | 76.83 | 62.74 | 53.51 | 57.83 | 54.73 | 52.27 |
| LR=0.001 Constant | 86.78 | 80.91 | 79.31 | 76.86 | 76.09 | 67.73 | 59.80 | 64.75 | 61.33 | 55.75 |
| LR=0.001 Cosine$^*$ | 86.94 | 80.83 | 79.11 | 77.50 | 74.55 | 70.03 | 61.22 | 66.56 | 63.15 | 55.15 |

Table 7: Effect of statistical modeling. We present the final average accuracy (FAA), cumulative average accuracy (CAA) and final forgetting measure (FFM) with $\pm$ standard deviation under different pre-trained models (PTM), over three runs of different random seeds and task splits.

| PTM | Method | Split CIFAR-100 | | | Split ImageNet-R | | |
|---|---|---|---|---|---|---|---|
| | | FAA ($\uparrow$) | CAA ($\uparrow$) | FFM ($\downarrow$) | FAA ($\uparrow$) | CAA ($\uparrow$) | FFM ($\downarrow$) |
| Sup-21K | Covariance | 92.61 $\pm$0.28 | 94.03 $\pm$0.01 | 3.16 $\pm$0.10 | 75.06 $\pm$0.12 | 76.60 $\pm$0.01 | 2.17 $\pm$0.19 |
| | Multi-Centroid | 92.92 $\pm$0.12 | 94.60 $\pm$0.06 | 0.86 $\pm$0.12 | 73.55 $\pm$0.21 | 75.93 $\pm$0.15 | 0.95 $\pm$0.02 |
| iBOT-21K | Covariance | 93.02 $\pm$0.15 | 94.56 $\pm$0.05 | 1.33 $\pm$0.24 | 70.83 $\pm$0.17 | 73.23 $\pm$0.08 | 2.46 $\pm$0.21 |
| | Multi-Centroid | 92.69 $\pm$0.10 | 94.69 $\pm$0.13 | 1.57 $\pm$0.24 | 70.63 $\pm$0.07 | 72.94 $\pm$0.06 | 1.31 $\pm$0.15 |
| iBOT-1K | Covariance | 93.48 $\pm$0.11 | 95.02 $\pm$0.01 | 1.00 $\pm$0.24 | 71.33 $\pm$0.21 | 73.62 $\pm$0.13 | 2.79 $\pm$0.26 |
| | Multi-Centroid | 91.78 $\pm$0.08 | 94.10 $\pm$0.10 | 1.45 $\pm$0.07 | 71.33 $\pm$0.29 | 74.38 $\pm$0.24 | 1.73 $\pm$0.37 |
| DINO-1K | Covariance | 92.51 $\pm$0.11 | 94.25 $\pm$0.01 | 0.99 $\pm$0.21 | 68.11 $\pm$0.18 | 71.70 $\pm$0.01 | 3.11 $\pm$0.17 |
| | Multi-Centroid | 90.06 $\pm$0.14 | 92.92 $\pm$0.19 | 2.13 $\pm$0.10 | 69.34 $\pm$0.16 | 72.35 $\pm$0.11 | 1.60 $\pm$0.16 |
| MoCo-1K | Covariance | 91.57 $\pm$0.20 | 93.70 $\pm$0.01 | 1.19 $\pm$0.18 | 63.77 $\pm$0.49 | 68.26 $\pm$0.01 | 3.57 $\pm$0.96 |
| | Multi-Centroid | 90.70 $\pm$0.02 | 93.23 $\pm$0.10 | 1.56 $\pm$0.12 | 63.05 $\pm$0.15 | 66.90 $\pm$0.13 | 1.78 $\pm$0.36 |

reserving fewer than 10 centroids (around 5 on average)[6] for each class achieves comparably strong performance in all cases (see Table 7), which ensures both efficiency and generality.

**5-Datasets**: In Table 8, we evaluate the continual learning performance on 5-Datasets for large inter-task differences. We keep the implementation of prompt architecture for all approaches the same as in Split CIFAR-100 and Split ImageNet-R. Then we perform an extensive grid search for learning rate, number of epochs and other applicable hyperparameters with their official or commonly-used codes. Under Sup-21K, we can essentially reproduce the results of L2P [41]. However, the reproduced results of DualPrompt [40] is lower than the reported one. We find that this issue is also discovered by other users and remains open in github. As the original paper of DualPrompt [40] has not described precisely the implementation for 5-Datasets, we speculate that this issue is possibly due to the use of different prompt architectures. A supporting evidence is that the reported results of DualPrompt [40] are similar to those of S-Prompt++, which is equivalent to replacing all task-sharing prompts in DualPrompt with task-specific prompts. Besides, the most recent CODA-Prompt [30] also performs poorly on this challenging benchmark. To ensure the strength of reproduced results, we have extensively searched the learning rate in $\{0.001, 0.0005\}$, the number of epochs in $\{5, 20, 40\}$ and the hyperparameter of orthogonality regularization in $\{0, 0.1, 0.01\}$, and presented the best performance. The inferiority of CODA-Prompt is possibly due to the excessive attentions to the previously-learned prompts, since the task distributions are clearly different and the old knowledge might interfere with the new one. To support this claim, we analyze the attentions of CODA-Prompt and observe a strong preference for previously-learned prompts (around $0.6 \sim 0.8$). Compared to all competitors, HiDe-Prompt achieves the strongest performance under different pre-training paradigms, consistent with the results on Split CIFAR-100 and Split ImageNet-R in Table 1. In particular, the FFM of HiDe-Prompt is almost zero, indicating its great success in overcoming catastrophic forgetting.

**Visualization of Representations**: We visualize the uninstructed and instructed representations with t-SNE, as shown in Fig. 5 for Sup-21K and Fig. 6 for iBOT-21K. Based on these results, we have the following analysis: (1) The uninstructed representations have shown single-peaked patterns in general, thanks to the use of adequate pre-training. This property allows them to be approximated with Gaussian distributions for preservation and recovery, and allows for correct prediction of task identity from them. (2) The instructed representations tend to be more compact and distinguishable, validating

---

[6]We use KNN to select multiple centroids (set to a maximum of 10) for each class of representations, where the average number of obtained centroids is usually around 5.

Table 8: Overall performance of continual learning on 5-Datasets.

| Baseline | FAA (↑) | | | | | FFM (↓) | | | | |
|---|---|---|---|---|---|---|---|---|---|---|
| | Sup-21K | iBOT-21K | iBOT-1K | DINO-1K | MoCo-1K | Sup-21K | iBOT-21K | iBOT-1K | DINO-1K | MoCo-1K |
| L2P [41] | 81.84 | 82.25 | 80.02 | 76.26 | 66.89 | 4.78 | 8.23 | 9.46 | 8.50 | 24.22 |
| DualPrompt [40] | 77.91 | 68.03 | 68.92 | 64.66 | 59.71 | 13.11 | 20.30 | 24.47 | 27.24 | 35.89 |
| S-Prompt++ [39] | 86.06 | 77.20 | 73.51 | 69.51 | 72.91 | 4.74 | 15.83 | 7.47 | 17.87 | 15.86 |
| CODA-Prompt [30] | 64.18 | 51.65 | 48.14 | 50.86 | 39.02 | 17.23 | 27.53 | 22.02 | 26.69 | 64.33 |
| HiDe-Prompt (Ours) | **93.83** | **94.88** | **93.89** | **93.50** | **93.28** | **0.44** | **0.09** | **0.07** | **0.04** | **0.14** |

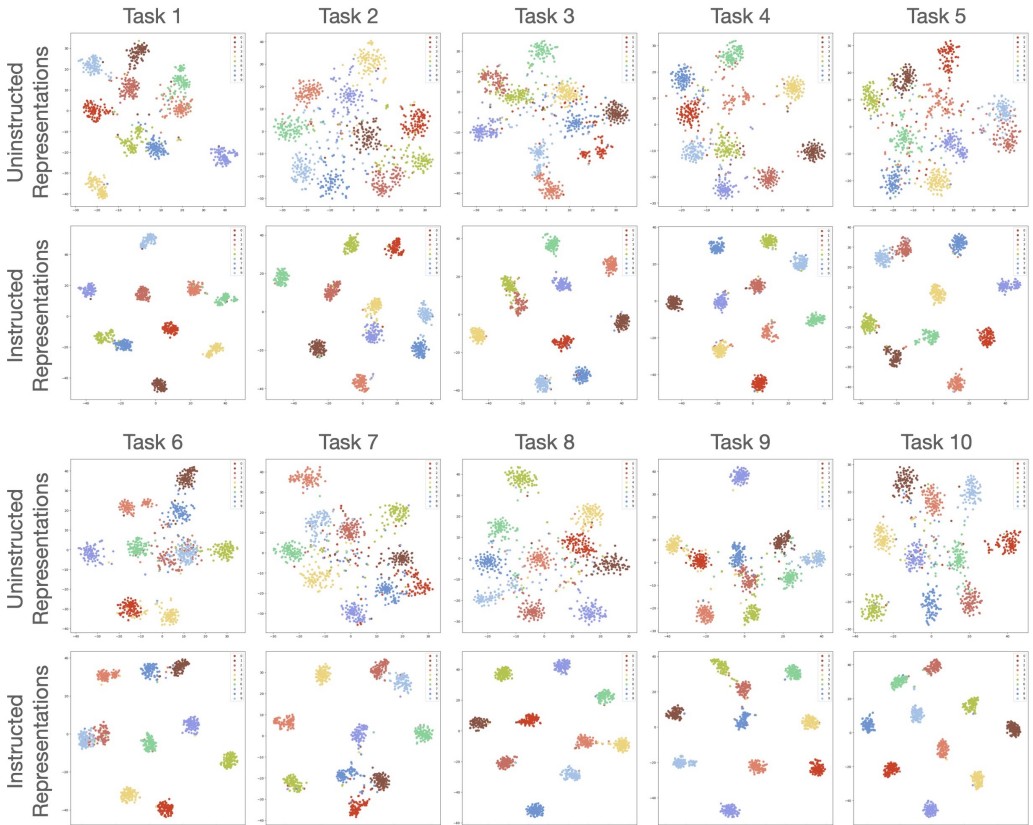

Figure 5: Visualization of uninstructed and instructed representations with t-SNE: Part I. Here we present the results of Split CIFAR-100 under Sup-21K. Each color represent a class.

the effectiveness of prompt-based continual learning. The degree of compactness and differentiation varies across pre-training paradigms, contributing to their performance differences. (3) The differences in compactness and differentiation between uninstructed and instructed representations suggest that the design of coarse-grained classification by task and fine-grained classification by class is reasonable for prompt-based continual learning, as do our theoretical analysis and the proposed approach.

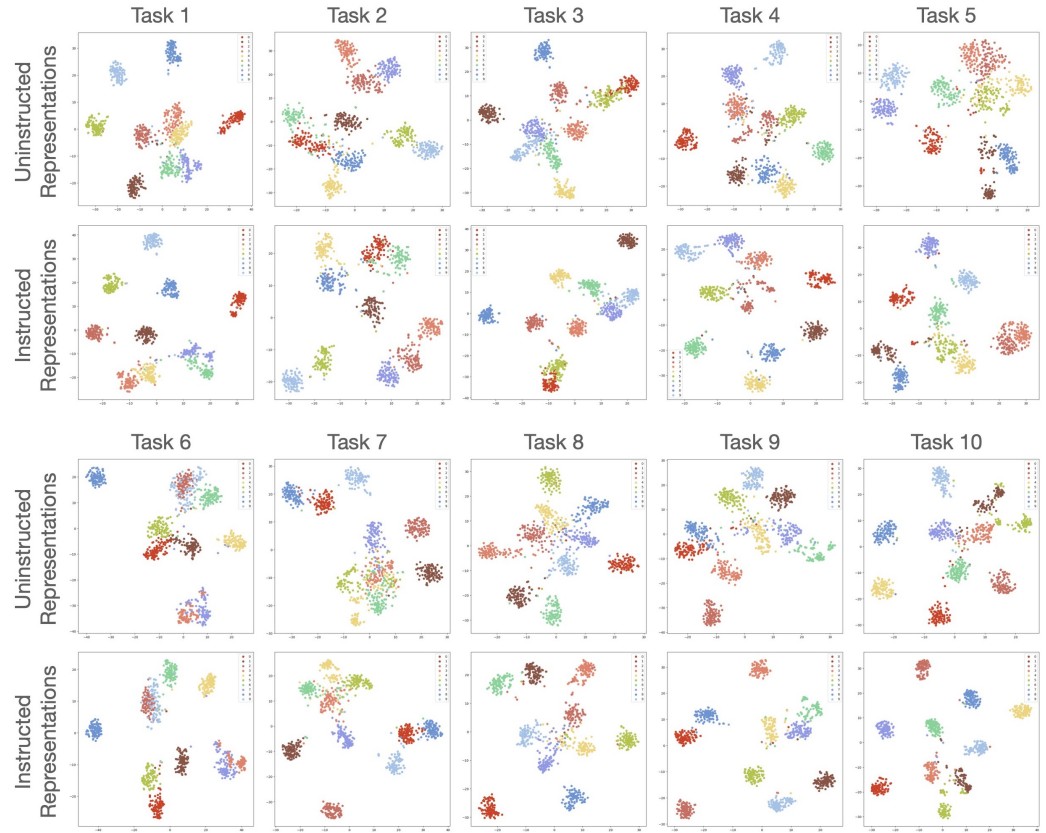

Figure 6: Visualization of uninstructed and instructed representations with t-SNE: Part II. Here we present the results of Split CIFAR-100 under iBOT-21K. Each color represent a class.

