# OpenReview forum: "Hierarchical Decomposition of Prompt-Based Continual Learning: Rethinking Obscured Sub-optimality"
_NeurIPS.cc/2023/Conference — NeurIPS 2023 spotlight_

### Official Review · Reviewer_UkQY · 2023-07-03

**Soundness:** 3 good
**Presentation:** 3 good
**Contribution:** 2 fair
**Rating:** 5
**Confidence:** 4

**Summary:**

This paper introduces a CIL approach that specifically addresses the use of pre-trained models. It is intriguing and significant to explore the most suitable CIL method for pre-trained models.



**Strengths:**

1. CIL and how to do CIL in pre-trained models are very important
2. The writing is smooth and very easy to follow

**Weaknesses:**

Excluding the assumption of a pre-trained model, this paper reduces to the existing research (as mentioned in line 233). Thus, I recommend conducting further analysis on CL for pre-trained models.

1. Considering the extensive exploration of pre-trained models in NLP, it is crucial for this paper to compare its approach with NLP continual learning baselines, e.g., [1,2,3].

2. I see no reason why the theory should be restricted to prompts alone. Other parameter-efficient tuning methods such as adapters, Lora, and Prefix can easily be utilized. The author should include a comparison with these methods (see 1 references).

3. In addition, I suggest comparing this paper's approach with TIL methods as well, which do not require TII. This would serve as an additional evaluation to assess the proposed method's performance in  WTP and TAP.

[1]: Achieving forgetting prevention and knowledge transfer in continual learning, NeurIPS 2021
[2]: Continual Pre-training of Language Models, ICLR 2023
[3]: Continual Learning of Natural Language Processing Tasks: A Survey. https://arxiv.org/abs/2211.12701

**Questions:**

See above

**Limitations:**

I think the discussion of limitations in Sec. 6 makes sense.

---

> ### Author Rebuttal · Authors · 2023-08-09
>
> Thank you for your positive feedback and insightful comments. Below, we provide a point-to-point response to these comments and summarize the corresponding revisions in final version.
>
> **Q1: Considering the extensive exploration of pre-trained models in NLP, it is crucial for this paper to compare its approach with NLP continual learning baselines, e.g., [1,2,3].**
>
> A1: Thank you for providing these excellent related work. CTR [1] focused on a task-incremental setting, inserting continual learning modules (typically capsule networks) into two locations of a BERT-like pre-trained model. In contrast, our work focuses on a class-incremental setting without the oracle of task identity at test time, and employs prompts as the ``inserted modules''.
> DAS [2] focused on continual pre-training of language models to improve their end-task performance, i.e., upstream continual learning. In comparison, our work focuses on improving the performance for all sequentially arrived tasks ever seen, i.e., downstream continual learning.
> The third reference [3] is a recent survey on continual learning of NLP tasks. It has summarized representative continual learning strategies, in particular advanced parameter-efficient fine-tuning techniques for continual learning with pre-training. As you mentioned, our work focuses on improving prompt-based methods in this direction. As shown in the responses to your Q2 and Reviewer Qf6e's Q5, our work could potentially be generalized to other PEFT techniques, serving as an important future work.
> We will add the related work and the above discussion in the final version.
>
> **Q2: I see no reason why the theory should be restricted to prompts alone. Other parameter-efficient tuning methods such as adapters, Lora, and Prefix can easily be utilized. The author should include a comparison with these methods (see 1 references).**
>
> A2: Thank you for your valuable suggestion. We agree that other parameter-efficient fine-tuning (PEFT) techniques could be utilized with our theoretical framework (please refer to the response to Reviewer Qf6e's Q5). All of these PEFT techniques can serve as the task-specific parameters in our HiDe-Prompt, differing mainly in their specific forms. Specifically, prompt-tuning and prefix-tuning both prepended a few parameters to input / hidden vectors, which have been discussed in line 118-124. Adapters inserted adaptive parameters between layers, while LoRa learned an additional low-rank matrix to approximate weight updates and added them to the backbone weights. In fact, we have attempted to implement LoRa in our method, which can still achieve considerable performance (e.g., 87.86%/92.24% on Split CIFAR-100 under Sup-21K/iBOT-21K pre-training). We are now working on implementing other PEFT techniques, which will be released in future work. We will add the above discussion in the final version.
>
> **Q3: In addition, I suggest comparing this paper's approach with TIL methods as well, which do not require TII. This would serve as an additional evaluation to assess the proposed method's performance in WTP and TAP.**
>
> A3: Thank you for your valuable suggestion. As the task identity is provided at test time, TIL can largely reduce the difficulty of continual learning compared to CIL and is usually implemented with task-specific output layers. Therefore, the TIL-version of our framework retains only WTP but removes both TII and TAP (see this conclusion and a theoretical proof in Appendix A.3). In other words, the continual learning problem for our approach becomes evaluating the performance of prompt-tuning itself, i.e., the ability of learning each task with task-specific prompts, without inter-task interference. Since most TIL methods focused on overcoming catastrophic forgetting, they do not constitute a direct comparison with the TIL-version of our approach in terms of motivation.
> Besides, we empirically validate that the TIL-version of our approach achieves sufficiently high performance (e.g., 97.83%/97.83% and 81.55%/80.37% on Split CIFAR-100 and Split ImageNet-R under Sup-21K/iBOT-21K pre-training, respectively), consistent with the above analysis. We will add the above discussion in the final version.
>
>
> [1] Achieving forgetting prevention and knowledge transfer in continual learning, NeurIPS 2021
>
> [2] Continual pre-training of language models, ICLR 2023
>
> [3] Continual learning of natural language processing tasks: A survey. https://arxiv.org/abs/2211.12701

---

> > ### Author Response · Authors · 2023-08-14
> > **Look forward to further feedback**
> >
> > We thank you again for the valuable and constructive comments. We hope you may find our response satisfactory and raise your rating accordingly.
> >
> > We are looking forward to hearing from you about any further feedback.
> >
> > Best,
> > Authors

---

> > > ### Comment · Reviewer_UkQY · 2023-08-19
> > >
> > > I appreciate the authors' response and have also read the reviews from other reviewers.
> > >
> > > In fact, I share a similar concern as reviewer Qf6e. Specifically, I'm unclear about the theoretical basis for the impact of pre-training. To me, the difference between pre-training or not lies in the initialization, and I do not fully understand how this could lead to a theoretical difference.

---

> > > > ### Author Response · Authors · 2023-08-20
> > > > **Author response**
> > > >
> > > > Thank you for your feedback. The objective of continual learning is to perform well on sequentially arrived tasks. There must be a theoretical basis for the huge performance differences between pre-trained and randomly-initialized weights in continual learning, especially for rehearsal-free CIL, which provides the initial motivation of our theoretical study.
> > > > An effective use of pre-trained knowledge for downstream continual learning also depends on specialized theories and methodologies.
> > > > Specifically, the randomly-initialized parameter set $\theta_0$ carries no information and is greatly different from the optimal solution. It needs to be substantially changed in continual learning and should not be took into account in theoretical analysis of the objective. In contrast, the pre-trained parameter set $\theta$ carries beneficial pre-trained knowledge for downstream tasks. If the pre-training is adequately strong, $\theta$ is already close to the optimal solution and only requires appropriate fine-tuning. Therefore, $\theta$ needs to be stabilized (usually frozen) in continual learning and should be considered in theoretical analysis. (An intuitive summary: the random initialization $\theta_0$ is overwritten, while the pre-trained initialization $\theta$ is preserved.)
> > > >
> > > > In our case, without considering the impact of pre-training, TAP = TII * WTP,  due to classifying the same input in the same semantic space, i.e., $\frac{\exp(h{\psi}(f(\boldsymbol{x}))[y])}{\sum_{j=1}^{t}\sum_{y' \in \mathcal{Y}j} \exp(h{\psi}(f(\boldsymbol{x}))[y'])}=\frac{\exp(\hat{h}{\omega}(f(\boldsymbol{x})[\bar{i}])}{\sum_{i=1}^{t} \exp(\hat{h}{\omega}(f(\boldsymbol{x})[i])}*\frac{\exp(h{\psi}(f(\boldsymbol{x}))[\bar{j}])}{\sum_{j \in \mathcal{Y}i} \exp(h_{\psi}(f(\boldsymbol{x}))[j])}$.
> > > >
> > > > With considering the impact of pre-training (the introduction of $\theta$), TAP != TII * WTP, because TII * WTP is to calculate $\frac{\exp(\hat{h}{\omega}(f(\boldsymbol{x})[\bar{i}])}{\sum_{i=1}^{t} \exp(\hat{h}{\omega}(f(\boldsymbol{x})[i])}$ $*\frac{\exp(h{\psi}(f(\boldsymbol{x}; \boldsymbol{p}{i}))[\bar{j}])}{\sum_{j \in \mathcal{Y}i} \exp(h{\psi}(f(\boldsymbol{x}; \boldsymbol{p}{i}))[j])}$, while TAP is to calculate $\frac{\exp(h{\psi}(f(\boldsymbol{x}; \boldsymbol{p}{i}))[y])}{\sum_{j=1}^{t}\sum_{y' \in \mathcal{Y}j} \exp(h{\psi}(f(\boldsymbol{x}; \boldsymbol{p}{i}))[y'])}$ (please refer to our latest response to Reviewer Qf6e for more details). In other words, the backbone parameters are frozen to stabilize pre-trained knowledge, with additional prompts $\boldsymbol{p}{i}$ to fine-tune the semantic space (i.e., representations). TAP is essentially classifying the instructed representation ($\boldsymbol{h} \in \mathcal{H}{i}^c$ in Eq. (12)), while TII * WTP is performed on uninstructed representation ($\hat{\boldsymbol{h}} \in \hat{\mathcal{H}}{i}^{c}$ in Eq. (11)). Targeting on different semantic spaces, TAP and TII * WTP are clearly different, i.e., TAP != TII * WTP. Strong empirical results also demonstrate their respective contributions to continual learning performance (see ablation study in Table 2).
> > > >
> > > > We thank you again for the valuable feedback. We will incorporate the above discussion in the final version to make our theoretical analysis clearer. If you have any further questions, please let us know.

---

### Official Review · Reviewer_6q9Z · 2023-07-04

**Soundness:** 3 good
**Presentation:** 3 good
**Contribution:** 4 excellent
**Rating:** 7
**Confidence:** 5

**Summary:**

This work provides a comprehensive analysis of state-of-the-art prompt-based approaches for continual learning with the use of pre-training. The authors empirically demonstrate a clear performance degradation of current strategies under realistic self-supervised pre-training and extensively analyze the exposed sub-optimality. The authors then provide an in-depth theoretical analysis of the continual learning objective in the context of pre-training, which can be decomposed into three hierarchical components, and propose an innovative approach to optimize them explicitly. Extensive experiments on various pre-training paradigms have demonstrated the clear advantages of the proposed method.

**Strengths:**

1. This paper provides a well-organized formulation of state-of-the-art prompt-based approaches, assessing them as a unified perspective.
2. The empirical analysis reveals the degraded performance of prompt-based approaches under self-supervised pre-training, which is an important issue for practical applications.
3. The theoretical analysis is very interesting. The hierarchical components are specific to continual learning in the context of pre-training and apply to the settings of task-/domain-/class-incremental learning.
4. The proposed method allows for explicit optimization of the continual learning objective through adaptively leveraging pre-training and prompt architectures. Experimental results demonstrate a significant improvement in continual learning performance under different pre-training paradigms.


**Weaknesses:**

1. Based on proofs in supplementary materials, is the notation $\bar{c}$ in Eq.(8) equal to $y$? Please check it.
2. Could the authors further discuss some concurrent related work such as PromptFusion [1] that exploits prompt and FSA [2] that exploits FiLM adapters for continual learning with pre-training.
[1] PromptFusion: Decoupling Stability and Plasticity for Continual Learning. arXiv preprint arXiv:2303.07223.
[2] First Session Adaptation: A Strong Replay-Free Baseline for Class-Incremental Learning. arXiv preprint arXiv:2303.13199.
3. Although experimental results have been sufficient enough, it would be better if some fine-grained datasets can be further analyzed, such as CUB.



**Questions:**

Some minor concerns remain to be addressed. Please refer to the Weakness.

**Limitations:**

The authors have properly discussed the limitations and potential negative societal impacts.

---

> ### Author Rebuttal · Authors · 2023-08-09
>
> Thank you for your positive feedback and insightful comments. Below, we provide a point-to-point response to these comments and summarize the corresponding revisions in final version.
>
> **Q1: Based on proofs in supplementary materials, is the notation $\bar c$ in Eq.(8) equal to $y$? Please check it.**
>
> A1: Yes, as you understand it, the notation $\bar c$ in Eq.(8) is equal to $y$, representing the ground truth label of $x$. For clarify, we will replace $\bar c$ with $y$ and add more explanations in the final version.
>
> **Q2: Could the authors further discuss some concurrent related work such as PromptFusion [1] that exploits prompt and FSA [2] that exploits FiLM adapters for continual learning with pre-training.**
>
> A2: Thank you for pointing out these excellent related work. We will include a discussion on them in the final version. Briefly, PromptFusion [1] employed two prompt-based models to optimize stability and plasticity, respectively, and combined their predictions in a weighted average. The two prompt-based models are constructed from a pre-trained ViT and an additional CLIP, and the combination of their predictions relies on a replay buffer of old training samples. In contrast, our work focuses on a rehearsal-free setting and only requires a pre-trained ViT, which is more resource-efficient and practical in applications.
> FSA [2] adapted a pre-trained backbone with FiLM adapters only in the first learning session and fixed it thereafter. In comparison, our work focuses on prompt-based techniques and adapts the backbone in all learning sessions, so as to accommodate subsequent changes in data distributions.
>
> **Q3: Although experimental results have been sufficient enough, it would be better if some fine-grained datasets can be further analyzed, such as CUB.**
>
> A3: Following your suggestion, we conducted an additional experiment on Split CUB-200-2011, i.e., a random split of its 200 classes into 10 tasks with 20 classes per task. The results are summarized as below:
>
> |   PTM    |     L2P    |  DualPrompt |  S-Prompt++ | CODA-Prompt | HiDe-Prompt |
> | -------- | ---------- | ----------- | ----------- | ----------- | ----------- |
> |  Sup-21K |    74.48   |    82.05    |    82.08    |    74.34    |  **86.56**  |
> | iBOT-21K |    44.29   |    41.31    |    42.73    |    47.79    |  **78.23**  |
>
> For continual learning of fine-grained classification tasks, the sub-optimality of representative prompt-based approaches is more clearly exposed under self-supervised pre-training (i.e., Sup-21K vs iBOT-21K). In comparison to these baselines, HiDe-Prompt (ours) achieves a substantial lead in performance (more than **30\%**). Through an extensive ablation study, we observe that this is largely due to the optimization of task-adaptive prediction (TAP) through replaying pseudo representations to adapt the final output layer, as the fine-grained classification generally requires a high degree of precision in classification outputs. Therefore, these results further demonstrate the importance of our empirical and theoretical contributions. We will add it in the final version.
>
>
> [1] PromptFusion: Decoupling Stability and Plasticity for Continual Learning. arXiv preprint arXiv:2303.07223.
>
> [2] First Session Adaptation: A Strong Replay-Free Baseline for Class-Incremental Learning. arXiv preprint arXiv:2303.13199.

---

> > ### Comment · Reviewer_6q9Z · 2023-08-18
> > **Thank you for your rebuttal.**
> >
> > I appreciate the authors’ effort to fully address my concerns. I have also read the rebuttal to other reviewers. The generality of the proposed method to other PEFT strategies (e.g., Lora) for continual learning is impressive. I believe this work could be influential in continual learning since efficient fine-tuning is very important for avoiding forgetting.

---

> > > ### Author Response · Authors · 2023-08-18
> > > **Thank you**
> > >
> > > We are happy to know that the reviewer recognizes the contribution and broad impact of our work. We also appreciate the positive feedback and strong support.
> > >
> > > Best,
> > > Authors

---

### Official Review · Reviewer_Qf6e · 2023-07-06

**Soundness:** 3 good
**Presentation:** 3 good
**Contribution:** 3 good
**Rating:** 5
**Confidence:** 3

**Summary:**

This paper studies application of prompts in pre-trained models for continual learning. Building upon the problem of [1], the authors introduce the Task-Adaptive Prediction (TAP) for the CIL problem using pre-trained networks. They demonstrate that a good TAP, WTP, and TII performances are necessary and sufficient for a good CIL model.
The authors emphasize the significance of these factors, especially in the context of continual learning with pre-trained self-supervised model.

[1] Kim et al., Theoretical study on solving continual learning. NeurIPS 2022

**Strengths:**

- The paper is easy to follow and the proposed approach is well motivated
- The paper has provided theoretical insights
- The proposed method is much stronger than the existing baselines


**Weaknesses:**

- The method heavily relies on pre-trained models trained with ImageNet. This can be an issue, especially for supervised pre-training, since ImageNet already contains classes similar or identical to classes used for CL and there could be information leak from the pre-training classes to continual learning classes [2].
- The theoretical contribution is somewhat ambiguous since the decomposition and theorem 1 and 2 are similar to [1].

[2] Kim et al., A multi-head model for continual learning via out-of-distribution replay. CoLLAs, 2022.

**Questions:**

- I couldn’t fully-understand why it’s necessary to have TAP in the CL problem using pre-trained networks, considering that it wasn’t required when training from scratch [1].
- I am curious about the performance of the proposed method when the feature extractor is pre-trained using only the classes that are dissimilar to CL classes as [2]
- Is the theoretical analysis only relevant to the prompt-based method?

**Limitations:**

Refer to Weaknesses and Questions

---

> ### Author Rebuttal · Authors · 2023-08-09
>
> Thank you for your positive feedback and insightful comments. Below, we provide a point-to-point response to these comments and summarize the corresponding revisions in final version.
>
> **Q1: The method heavily relies on pre-trained models trained with ImageNet. This can be an issue, especially for supervised pre-training, since ImageNet already contains classes similar or identical to classes used for CL and there could be information leak from the pre-training classes to continual learning classes [2].**
>
> A1: In this work, we follow the pre-training dataset from previous work of prompt-based continual learning and focus on the impact of pre-training paradigms (especially *self-supervised pre-training*). Following your suggestion in Q4, we perform an additional experiment of using only the classes that are dissimilar to downstream CL classes for pre-training [2]. In this experimental setup, our approach achieves a more significant lead in performance (19.05\% on Split CIFAR-100), which is further detailed in our response to Q4. We will add these results in the final version.
>
> **Q2: The theoretical contribution is somewhat ambiguous since the decomposition and theorem 1 and 2 are similar to [1].**
>
> A2: In this work, we focus on continual learning with pre-training, especially prompt-based continual learning that has recently received significant attention in this direction. Our theoretical contribution lies in the context of **pre-training**, where we demonstrate the necessary conditions to achieve good continual learning performance. This is clearly different from the theorem in [1] about continual learning from scratch. Specifically, the major differences between ours and [1] include: (1) The condition is different. We formulate the decomposed components (i.e., TII, WTP and TII) as $\theta$-conditional probabilities (i.e., $P(\boldsymbol{x} \in \mathcal{X}{i}|\mathcal{D},\theta)$,
> $P(\boldsymbol{x} \in \mathcal{X}{i,j}|\boldsymbol{x} \in \mathcal{X}{i},\mathcal{D},\theta)$ and $ P(\boldsymbol{x} \in \mathcal{X}^{c}|\mathcal{D},\theta)$), where $\theta$ captures the pre-trained knowledge while not considered in [1]. (2) Due to the additional introduction of $\theta$, the necessary conditions to achieve good continual learning performance are different from that in [1]. Beside WTP and TII, TAP is especially necessary in the CL problem using pre-training (please refer to A3 for more details). We will add more explanations to make it clearer.
>
> **Q3: Why it’s necessary to have TAP in the CL problem using pre-trained networks, considering that it wasn’t required when training from scratch [1].**
>
> A3: As stated in A2, from the theoretical perspective, TAP is equivalent to TII with WTP when training from scratch. That means $ P(\boldsymbol{x} \in \mathcal{X}^{y}|\mathcal{D},\theta) = P(\boldsymbol{x} \in \mathcal{X}{\bar{i}}|\mathcal{D},\theta)P(\boldsymbol{x} \in \mathcal{X}{\bar{i},\bar{j}}|\boldsymbol{x} \in \mathcal{X}{\bar{i}},\mathcal{D},\theta) $, corresponding to $\delta +\epsilon = \eta$ in our Theorem 1. Therefore, it is unnecessary to have TAP in CIL when training from scratch, which is consistent to [1] as discussed in line 231-233 of our manuscript. On the other hand, when using pre-training with the hierarchical framework as shown in Fig. 3, TAP and TII with WTP are formulated as two different prediction problems where $ P(\boldsymbol{x} \in \mathcal{X}^{y}|\mathcal{D},\theta) \neq P(\boldsymbol{x} \in \mathcal{X}{\bar{i}}|\mathcal{D},\theta)P(\boldsymbol{x} \in \mathcal{X}{\bar{i},\bar{j}}|\boldsymbol{x} \in \mathcal{X}{\bar{i}},\mathcal{D},\theta) $. Thus, the final performance of having TAP in CIL (i.e., $\max [P(\boldsymbol{x} \in \mathcal{X}{\bar{i},\bar{j}}|\mathcal{D},\theta),P(\boldsymbol{x} \in \mathcal{X}^{y}|\mathcal{D},\theta)]$) can outperform that without TAP (i.e., $P(\boldsymbol{x} \in \mathcal{X}{\bar{i},\bar{j}}|\mathcal{D},\theta)$). These results theoretically demonstrate that the use of pre-training can indeed improve continual learning compared to training from scratch. We will make it clearer.
>
> **Q4: The performance of the proposed method when the feature extractor is pre-trained using only the classes that are dissimilar to CL classes as [2].**
>
> A4: Following the setup in [2], we use a pre-trained checkpoint of the ImageNet subset (with 389 similar classes removed) for continual learning of Split CIFAR-100. The final average accuracy (FAA) of S-Prompt++, CODA-Prompt and our approach is 69.00%, 65.07% and 88.05%, respectively. As can be seen, the performance lead of our approach becomes significantly much larger in this experimental setup, thanks to the explicit optimization of hierarchical components to overcome sub-optimal aspects in prompt-based continual learning.
>
> **Q5: Is the theoretical analysis only relevant to the prompt-based method?**
>
> A5: Indeed, our theoretical analysis could be extended as a general framework of parameter-efficient fine-tuning (PEFT) for continual learning with pre-training. Specifically, based on our theoretical analysis, the objective of continual learning is achieved by three components: (1) optimization of WTP with task-specific parameters, (2) optimization of TII with uninstructed representations, and (3) optimization of TAP with instructed representations. Mainstream PEFT techniques are applicable to this framework in general. The differences lie only in the form of task-specific parameters used in (1), which could be prompt, adapter, LoRA, FiLM, etc. In fact, we have attempted to implement LoRa in our method, which can still achieve considerable performance (e.g., 87.86%/92.24% on Split CIFAR-100 under Sup-21K/iBOT-21K pre-training). Since the major focus of this paper is to improve prompt-based continual learning, which is one of the most active technical routes in this direction, we leave the extension of other PEFT techniques as well as an empirical comparison to further work. We will add it in the final version.

---

> > ### Author Response · Authors · 2023-08-14
> > **Look forward to further feedback**
> >
> > We thank you again for the valuable and constructive comments. We hope you may find our response satisfactory and raise your rating accordingly.
> >
> > We are looking forward to hearing from you about any further feedback.
> >
> > Best,
> > Authors

---

> > > ### Comment · Reviewer_Qf6e · 2023-08-18
> > > **Response to the author comments**
> > >
> > > Thank you for providing detailed responses. However, I still do not fully understand the theoretical part.
> > >
> > > With current manuscript, I don't see a significant theoretical contribution of this paper. The outcomes seem to be a direct extension of those presented in [1] with one additional consideration, $max\\{CIL, TAP\\}$. Moreover, I have noticed that the notations, necessary and sufficient conditions, theorem 1, and proofs are very similar to [1].
> > >
> > > Regarding the author's comment A2, I don't see the distinction made by having the $\theta$ condition in the probability. The probability decomposition as well as all the theoretical results in [1] remain the same with or without $\theta$.
> > >
> > > Regarding the author's comment A3, what is the difference between the TAP probability $P(\mathcal{X}^y | D, \theta)$ and CIL probability $P(\mathcal{X}_{i, j} | D, \theta)$? Why TAP = TII * WTP in [1] becomes TAP != TII * WTP in this pre-training setting?
> > >
> > > Misc. Figure 3 is presented, but it is not referenced in the paper.
> > >
> > > I am looking forward to your comment!

---

> > > > ### Author Response · Authors · 2023-08-19
> > > > **Author response (part I)**
> > > >
> > > > Thanks for your feedback. Here we explain your additional questions about our theoretical part as below (we reorganize the order of questions and answers for clarity).
> > > >
> > > > **Q2-1: With current manuscript, I don't see a significant theoretical contribution of this paper. The outcomes seem to be a direct extension of those presented in [1] with one additional consideration, max[CIL,TAP]. Moreover, I have noticed that the notations, necessary and sufficient conditions, theorem 1, and proofs are very similar to [1].**
> > > >
> > > > A2-1: Our main contribution in the theoretical part is to demonstrate the importance of WTP, TII and **TAP** for continual learning with **pre-training**. This is distinguished from the main conclusion in [1] due to the introduction of pre-trained knowledge, with the necessity of all three components empirically justified in Table 2. As the additional consideration, TAP is performed specifically on the **instructed representation space**, which is highly dependent on the pre-training backbone (i.e., $\theta$), aiming to improve predictions by coupling it with TII and WTP. Since [1] is a pioneering theoretical study decomposing the objective of continual learning from scratch, we build on its notation and analysis for the shared part (i.e., WTP and TII), in order to provide an explicit comparison and demonstrate clearly our own contribution (i.e., TAP in response to the impact of pre-training). In fact, the differences and connections of [1] and ours have been explicitly discussed in our paper (see line 231-234).
> > > >
> > > > As summarized in the contribution part of the paper (line 50-56), our theoretical results provide strong insights to analyze the sub-optimality of representative methods for prompt-based continual learning and to address the sub-optimality with an innovative approach. Therefore, we respectfully argue that our theoretical contribution is not in isolation but highly synergistic with the empirical findings and performance improvements, as acknowledged by Reviewer pdqM, 6q9Z and tgkY. These results can potentially provide a unified perspective for parameter efficient fine-tuning in continual learning with pre-training, as shown in the response to your Q5 in the Rebuttal page.
> > > >
> > > > **Q2-2: Regarding the author's comment A3, what is the difference between the TAP probability and CIL probability? Why TAP = TII * WTP in [1] becomes TAP != TII * WTP in this pre-training setting?**
> > > >
> > > > A2-2: In this paper, the TAP probability $ P(\boldsymbol{x} \in \mathcal{X}^{y}|\mathcal{D},\theta)$ is a categorical distribution over all observed classes $\bigcup_{k=1}^{t} \mathcal{Y}k$. The typical CIL probability $P(\boldsymbol{x} \in \mathcal{X}{\bar{i},\bar{j}}|\mathcal{D},\theta)$ can be further divided into TII * WTP where $\theta$ carries the pre-trained knowledge. Without considering pre-training as in [1], TAP = TII * WTP, i.e., $ P(\boldsymbol{x} \in \mathcal{X}^{y}|\mathcal{D})=P(\boldsymbol{x} \in \mathcal{X}{\bar{i},\bar{j}}|\mathcal{D})$, due to classifying the same input in the same semantic space, i.e., $\frac{\exp(h{\psi}(f(\boldsymbol{x}))[y])}{\sum_{j=1}^{t}\sum_{y' \in \mathcal{Y}j} \exp(h{\psi}(f(\boldsymbol{x}))[y'])}=\frac{\exp(\hat{h}{\omega}(f(\boldsymbol{x})[\bar{i}])}{\sum_{i=1}^{t} \exp(\hat{h}{\omega}(f(\boldsymbol{x})[i])}*\frac{\exp(h{\psi}(f(\boldsymbol{x}))[\bar{j}])}{\sum_{j \in \mathcal{Y}i} \exp(h_{\psi}(f(\boldsymbol{x}))[j])}$.
> > > >
> > > > In contrast, with considering pre-training the network $f$ with parameter set $\theta$ in our paper, TAP != TII * WTP, i.e., $ P(\boldsymbol{x} \in \mathcal{X}^{y}|\mathcal{D},\theta)\neq P(\boldsymbol{x} \in \mathcal{X}{\bar{i},\bar{j}}|\mathcal{D},\theta)$. This is because, TII * WTP is to calculate $\frac{\exp(\hat{h}{\omega}(f(\boldsymbol{x})[\bar{i}])}{\sum_{i=1}^{t} \exp(\hat{h}{\omega}(f(\boldsymbol{x})[i])}$ $*\frac{\exp(h{\psi}(f(\boldsymbol{x}; \boldsymbol{p}{i}))[\bar{j}])}{\sum_{j \in \mathcal{Y}i} \exp(h{\psi}(f(\boldsymbol{x}; \boldsymbol{p}{i}))[j])}$, while TAP is to calculate $\frac{\exp(h{\psi}(f(\boldsymbol{x}; \boldsymbol{p}{i}))[y])}{\sum_{j=1}^{t}\sum_{y' \in \mathcal{Y}j} \exp(h{\psi}(f(\boldsymbol{x}; \boldsymbol{p}{i}))[y'])}$. In other words, with prompts $\boldsymbol{p}{i}$ adapting the feature space (see Fig. 3), TAP is essentially classifying the instructed representation ($\boldsymbol{h} \in \mathcal{H}^c_{i}$ in Eq. (12)), while TII * WTP is performed on uninstructed representation ($\hat{\boldsymbol{h}} \in \hat{\mathcal{H}}_{i}^{c}$ in Eq. (11)).

---

> > > > ### Author Response · Authors · 2023-08-19
> > > > **Author response (part II)**
> > > >
> > > > **Q2-3: Regarding the author's comment A2, I don't see the distinction made by having the $\theta$ condition in the probability. The probability decomposition as well as all the theoretical results in [1] remain the same with or without $\theta$.**
> > > >
> > > > A2-3: It is worth noting that when having $\theta$ condition in the probability of TII, WTP and TAP, prompting could change the original input space (i.e., the uninstructed representation) as the instructed representation.
> > > > Since TAP classifies the instructed representation over all observed classes, it is not equivalent to TII * WTP that is performed on uninstructed representation (please refer to A2-2 above for more details). As a result, the probability decomposition in [1] (i.e., TII * WTP) will be evolved into our multi-objective decomposition max[TII * WTP, TAP], so as to improve continual prediction. A detailed proof of these results is included in Appendix A, which has been double checked to ensure solidity.
> > > >
> > > > We thank you again for these valuable feedback. We will add the discussions in A2-1, A2-2 and A2-3 to make our theoretical results clearer.
> > > >
> > > > **Q2-4: Misc. Figure 3 is presented, but it is not referenced in the paper.**
> > > >
> > > > A2-4: Thank you for pointing out this. Fig. 3 is an illustration of our method (HiDe-Prompt) coupled with Sec. 4.2. We will clarify it in line 241-242.

---

### Official Review · Reviewer_tgkY · 2023-07-08

**Soundness:** 3 good
**Presentation:** 4 excellent
**Contribution:** 3 good
**Rating:** 7
**Confidence:** 5

**Summary:**

The authors provide strong empirical analysis on existing "prompting for continual learning" papers. They propose a new hierarchical prompting method that includes several components which take advantage of unstructured data representations. The authors not only propose an interesting approach with SOTA performance, but they have very detailed experiments using several pre-training backbones and datasets.

**Strengths:**

In general, I feel that for this subject area (which I am sure will have many neurips submissions), this paper is overall high quality.
1) I appreciate another "prompting for continual learning" paper that starts with insights and findings into existing methods, laying an intuitive foundation for the proposed approach.
2) With good intuitive and theoretical backing that seems to reasonable, the method has great performance gains over SOTA.
3) Nice, detailed analysis and ablation study. I checked carefully to see if maybe there was a "hidden" component that does most of the heavy lifting of the method, but the contribution components seem to all be impactful in some way.
4) I am incredibly thankful to see the experiments conducted on several different backbones. This code and framework will be a fantastic contribution on its own, along with the analysis, before even considering the method. As someone who has experience in this problem setting, I think open-source code for this project will be greatly appreciated.

**Weaknesses:**

1) Task-id prediction for class-incremental learning on random splits of classes (i.e., where there is no actual task structure), seems a red flag to me that the pre-trained backbone is too strong for the proposed task datasets, and that it is actually taking advantage of the unfair backbone in order to "circumvent" class-incremental learning, instead posing it as the much easier task-incremental learning problem. However, this weakness also exists in the competing methods, and the strengths of the paper out-weigh this weakness imo.
2) Figure 2 c and d is very confusing. Which method is this for? If you are making strong claims about task specificity, should it not be done for all methods?
3) This is not relevant to your findings, and I am not requesting any experiments, but Coda-prompt might perform better at a lower constant learning rate if the high, constant learning rate is hurting its performance. Learning rate decay is also not a "trick", either - it is a valid hyperparameter choice and important for many representation learning methods.

**Questions:**

a) See weakness 2
b) what is the computation cost of your methods compared to the baselines? I feel that even with a strong training time cost, I am happy to vote for paper acceptance. Just want to ensure transparency.

**Limitations:**

Discussed reasonable limitations - there is no obvious potential negative societal impact

---

> ### Author Rebuttal · Authors · 2023-08-09
>
> Thank you for your positive feedback and insightful comments. Below, we provide a point-to-point response to these comments and summarize the corresponding revisions in final version. We are pleased that our implementation code is appreciated. It will be published after acceptance.
>
> **Q1: Task-id prediction for class-incremental learning on random splits of classes...it is actually taking advantage of the unfair backbone in order to "circumvent" class-incremental learning, instead posing it as the much easier task-incremental learning problem.**
>
> A1: We agree that the pre-trained backbones considered in this work as well as most prompt-based continual learning papers are adequately strong for downstream tasks. In fact, this experimental setup is **complementary** to the commonly-used protocol of class-incremental learning (CIL), which learns incremental classes from scratch or relatively weak pre-training (i.e., learning half of the classes in the first stage). As large-scale pre-training has proven to greatly facilitate a variety of downstream tasks, how to exploit it effectively in continual learning is of increasing interest. Our work demonstrates that CIL has distinctive results with relatively strong pre-training (e.g., more tolerance to the errors in task-identity inference), which extends previous explorations of CIL.
>
> On the other hand, the CIL in our experiments remains clearly **different** from TIL. As shown in the response to Reviewer UkQY's Q3, the performance of TIL is remarkably better than that of CIL. This is because the access to task identity enables TIL to avoid inter-task interference via multi-head output layers, while CIL usually requires a well-adapted single-head output layer to avoid errors from predicting an incorrect task identity. In fact, the relationship between CIL, TIL and pre-training has been shown in our theoretical analysis, i.e., a cross-talk between within-task prediction (WTP), task-identity inference (TII) and task-adaptive prediction (TAP). Removing the significant impact of pre-training would degenerate our framework into regular CIL, i.e., only WTP and TII remain (see line 231-234). As for TIL, the task identity is provided for multi-head evaluation and thus only WTP remains (see Appendix A.3 and Reviewer UkQY's A3).
>
> Besides, we have evaluated our approach on 5-Dataset, a benchmark with **actual task structure** (see line 287-289), and it can still achieve a significant performance lead (see Appendix Table 6 and line 602-624).
>
> We will add the above discussion in the final version.
>
> **Q2: Figure 2 c and d is very confusing. Which method is this for? If you are making strong claims about task specificity, should it not be done for all methods?**
>
> A2: In Fig. 2c, we analyze the instructed representations of task-specific prompts, which are identical to the prompt architecture of S-Prompt++ (see line 181-182, 164-168). In Fig. 2d, we analyze the ability of uninstructed representations to predict task identity, corresponding to the same method in Fig. 2c.
> For all the prompt-based methods considered in Fig. 1, since the training set of each task is provided sequentially, the optimizable parameters for each task inevitably acquire task-specific knowledge and the corresponding knowledge needs to be invoked correctly at test time. However, due to the complexity of their prompt architectures (e.g., L2P, DualPrompt and CODA-Prompt all reuse some prompt parameters optimized for previous tasks), task specificity is difficult to demonstrate explicitly in experiments as it is with S-Prompt++. Therefore, we construct such a demo experiment to analyze task specificity in prompt-based continual learning.
> In fact, the objective of all methods is tantamount to optimizing the probability distribution on the left side of Eq.(5), which can be decomposed into the two probabilities on the right side corresponding to Fig. 2c and 2d, respectively. Therefore, our demo experiment is representative for prompt-based methods from a theoretical perspective. We will add more descriptions and explanations to make it clearer.
>
> **Q3: Coda-prompt might perform better at a lower constant learning rate if the high constant learning rate is hurting its performance. Learning rate decay is also not a "trick", either - it is a valid hyperparameter choice and important for many representation learning methods.**
>
> A3: The main results of CODA-Prompt (i.e., Table 1) are produced from **using a lower learning rate with cosine decay** (see line 300), which is the same as its original paper and indeed performs better than a higher constant learning rate (see Fig. 2a, 2b and Appendix Table 5). We have also performed some experiments to validate that using such a lower learning rate with cosine decay is slightly better or comparable to using a lower constant learning rate (see Appendix Table 5). For example, the final average accuracy (FAA) of CODA-Prompt with a learning rate of 0.005 (constant), 0.001 (constant) and 0.001 (cosine decay) on Split CIFAR-100 under Sup-21K pre-training is 85.92%, 86.78%, and 86.94%, respectively. We agree that the learning rate decay is a valid hyperparameter choice for deep learning methods. We will modify our claim appropriately and add more explanations to avoid potential misunderstanding.
>
> **Q4: What is the computation cost of your methods compared to the baselines?**
>
> A4: Using the same single-card A100 GPU, the training times of L2P, DualPrompt, S-Prompt++, CODA-Prompt and our approach are 0.55h, 2.00h, 2.01h, 2.08h, and 2.80h on Split CIFAR-100, respectively. We observe that L2P requires a much smaller epoch number for convergence but performs the worst in general. Compared to other strong baselines, the computation cost of our approach is comparable in order of magnitude. In practice, we include parallel training in our implementation code, which can largely reduce the training time. We will add the above results in the final version.

---

> > ### Comment · Reviewer_tgkY · 2023-08-15
> > **Thank you for the rebuttal**
> >
> > I would like to thank the authors for answering my concerns and questions. I will remain at a score of 7 and recommend this paper be accepted to NeurIPS 2023. Congratulations on the great work.

---

> > > ### Author Response · Authors · 2023-08-15
> > > **Thank you**
> > >
> > > Thank you so much for your support. We appreciate it.
> > >
> > > Best, Authors

---

### Official Review · Reviewer_pdqM · 2023-07-09

**Soundness:** 3 good
**Presentation:** 3 good
**Contribution:** 3 good
**Rating:** 8
**Confidence:** 5

**Summary:**

The paper suggests a new prompt-based continual learning method by leveraging combinatorial objectives with *within task prediction*, *task-identity inference*, and *task-adaptive prediction*. The paper first summarizes recent prompt-based continual learning techniques and demonstrates their unstable performance according to the pre-trained backbones (particularly, initializing with self-supervised representation degrades their performance significantly). And then, proposes a new regularization technique containing core components, including ensemble prompting, contrastive regularization, and TAP loss.

**Strengths:**

The paper provides a comprehensive analysis and theoretical proof of the motivation. And the suggested idea is reasonable, and the methodological design is also clear. In experiments, the proposed method consistently surpasses strong baselines in terms of conventionally used continual learning metrics under multiple pre-trained backbones, and its improvement and stability are a bit impressive.
Further, they provide the necessary ablation study and analysis in the main paper and appendix.

**Weaknesses:**

The effect of the prompt ensemble in the method is not discussed. And the different behavior and impact of un-/instructed representation for continual learning are partially discussed in Figure 5, but I recommend a more comprehensive and detailed discussion/analysis.

**Questions:**

N/A

**Limitations:**

Please see the weakness.

---

> ### Author Rebuttal · Authors · 2023-08-09
>
> Thank you for your positive feedback and insightful comments. Below, we provide a point-to-point response to these comments and summarize the corresponding revisions in final version.
>
> **Q1: The effect of the prompt ensemble in the method is not discussed.**
>
> A1: We would respectfully point out that the effect of the prompt ensemble has been discussed in experiments, referred to as "WTP" in Table 2 and Fig. 4a. As shown in line 248-255, the prompt ensemble is proposed to learn each task more effectively with the architecture of task-specific prompts, so as to improve within-task prediction (WTP). In comparison to the "naive architecture" that employs only task-specific prompts, our prompt ensemble strategy can largely improve the performance of learning each new task (Fig. 4a) and thus improve the average performance of all tasks (Table 2), consistent with the motivation of our design. We will add more explanations to make it clearer.
>
> **Q2: The different behavior and impact of un-/instructed representation for continual learning are partially discussed in Figure 5, but I recommend a more comprehensive and detailed discussion/analysis.**
>
> A2: Thank you for your valuable suggestion. We have performed an extensive visualization of un-/instructed representations in terms of continual learning methods and pre-training paradigms. Fig. 5 reports the representations of our approach under iBOT-21K pre-training, and the results of Sup-21K are further included in the rebuttal PDF. From these results, we provide a more comprehensive analysis as below:
>
> (1) The uninstructed representations have shown single-peaked patterns in general, thanks to the use of adequate pre-training. This property allows them to be approximated as Gaussian distributions for preservation and recovery, and allows for correct prediction of task identity from them as shown in Fig. 4b.
>
> (2) The instructed representations tend to be more compact and distinguishable, validating the effectiveness of prompt-based continual learning. The degree of compactness and differentiation varies across continual learning methods and pre-training paradigms, contributing to their performance differences in Table 1.
>
> (3) The differences in compactness and differentiation between un-/instructed representations suggest that the design of coarse-grained classification by task and fine-grained classification by class is reasonable for prompt-based continual learning, as do our theoretical analysis and the proposed method.
>
> We will add the above discussion with more visualization results in the final version.

---

> > ### Comment · Reviewer_pdqM · 2023-08-14
> > **Reviewer response**
> >
> > Thank you so much! After sincerely reading the author's rebuttal, all concerns are solved! I'm happy to keep my initial score. Congratulation on publishing strong work in the continual learning field!

---

> > > ### Author Response · Authors · 2023-08-14
> > > **Thank you**
> > >
> > > We thank the reviewer for finding our response satisfactory and are happy to know the very positive rating.
> > >
> > > Best,
> > > Authors

---

### Author Rebuttal · Authors · 2023-08-09

We thank all reviewers for their great efforts and constructive comments, which help us to further improve the manuscript. We have tried our best to address these comments with additional experiments, explanations and discussions. Please let us know if you have any further questions.

---

### Decision · Program_Chairs · 2023-09-21

**Decision:**

Accept (spotlight)

**Comment:**

This paper provides theoretical insight and an empirical method to prompt-based continual learning utilizing self-supervised pre-training. Specifically, a hierarchical decomposition of the continual learning objective is analyzed, yielding within-task prediction, task-identity inference, and task-adaptive prediction as components. Empirically, an ensemble of task-specific prompts and contrastive regularization is employed to optimize the identified hierarchical components. Strong results are shown across a range of architectures and datasets, demonstrating state of art results.

  All of the reviewers appreciated the work, including the comprehensive insights and analysis leading to the method (pdqM, tgkY), strong methodology (pdqM), and strong empirical performance and ablations (pdqM, tgkY). All of the reviewers had positive scores, with some questions/concerns about the need and success for task-id prediction, lack of fine-grained datasets, and reliance on a pre-trained ImageNet model. The borderline reviewers also asked about the significance of the theoretical contribution, since the role of the pre-training part in the theory is unclear compared to theoretical analysis in prior works.

  The authors provided a thorough response, including new experiments (e.g. fine-grained datasets) and in particular engaged with the reviewers about difference in theoretical analysis with or without pretraining (as analyzed in prior theory). Overall, the rebuttals addressed the substantive parts of the concerns in my view. The setting of utilizing pre-trained self-supervised models is extremely important for continual learning, and in fact some of the discussions about applicability to other PEFT methods could even more significantly increase the impact of this paper. I recommend acceptance and encourage the authors to include a lot of the interesting outcomes of the discussion with reviewers.